# Well-Calibrated Regression Uncertainty
# in Medical Imaging with Deep Learning

**Max-Heinrich Laves[1]**                                    LAVES@IMES.UNI-HANNOVER.DE

**Sontje Ihler[1]**                                          IHLER@IMES.UNI-HANNOVER.DE

**Jacob F. Fast[1]**                                         FAST@IMES.UNI-HANNOVER.DE

**Lüder A. Kahrs[2,3]**                                      LAKAHRS@CS.TORONTO.EDU

**Tobias Ortmaier[1]**                                       ORTMAIER@IMES.UNI-HANNOVER.DE

[1]*Institute of Mechatronic Systems, Leibniz Universität Hannover, Hanover, Germany*

[2]*Centre for Image Guided Innovation and Therapeutic Intervention, The Hospital for Sick Children, Toronto, Canada*

[3]*Department of Mathematical and Computational Sciences, University of Toronto Mississauga, Mississauga, Canada*

## Abstract

The consideration of predictive uncertainty in medical imaging with deep learning is of utmost importance. We apply estimation of predictive uncertainty by variational Bayesian inference with Monte Carlo dropout to regression tasks and show why predictive uncertainty is systematically underestimated. We suggest using $\sigma$ *scaling* with a single scalar value; a simple, yet effective calibration method for both aleatoric and epistemic uncertainty. The performance of our approach is evaluated on a variety of common medical regression data sets using different state-of-the-art convolutional network architectures. In all experiments, $\sigma$ scaling is able to reliably recalibrate predictive uncertainty. It is easy to implement and maintains the accuracy. Well-calibrated uncertainty in regression allows robust rejection of unreliable predictions or detection of out-of-distribution samples. Our source code is available at: github.com/mlaves/well-calibrated-regression-uncertainty

**Keywords:** Bayesian approximation, variational inference

## 1. Introduction

For the task of regression, we aim to estimate a continuous target value $\boldsymbol{y} \in \mathbb{R}^d$ given an input image $\boldsymbol{x}$. Regression in medical imaging with deep learning has been applied to forensic age estimation from hand CT/MRI (Halabi et al., 2019; Štern et al., 2016), natural landmark localization (Payer et al., 2019), cell detection in histology (Xie et al., 2018), or instrument pose estimation (Gessert et al., 2018). By predicting the coordinates of object boundaries, segmentation can also be performed as a regression task. This has been done for segmentation of pulmonary nodules in CT (Messay et al., 2015), kidneys in ultrasound (Yin et al., 2020), or left ventricles in MRI (Tan et al., 2017). In registration of medical images, a continuous displacement field is predicted for each coordinate of $\boldsymbol{x}$, which has also recently been addressed by CNNs for regression (Dalca et al., 2019).

In medical imaging it is crucial to consider the predictive uncertainty of deep learning models. Bayesian neural networks (BNN) and their approximation provide mathematical

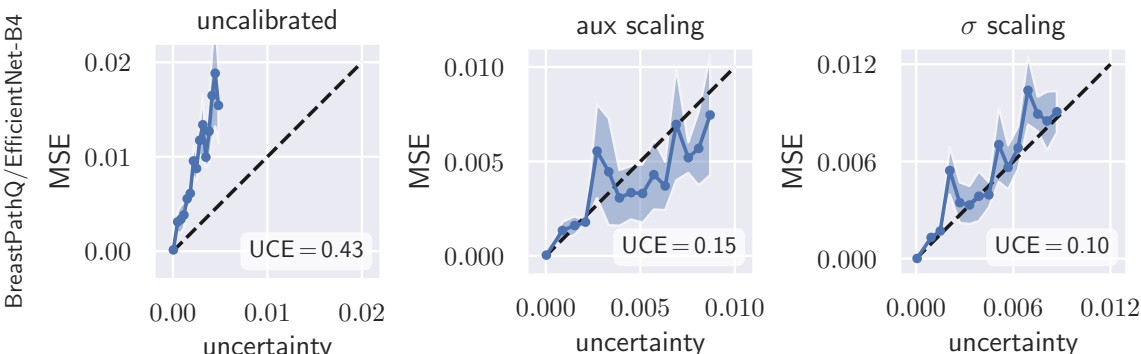

Figure 1: Calibration plots and uncertainty calibration error (UCE) for EfficientNet-B4 on BreastPathQ test set. Uncalibrated uncertainty is underestimated and does not correspond well with the model error (left). Uncertainty can be calibrated most effectively with $\sigma$ scaling (right). Solid lines show the mean and shaded areas show standard deviation from 5 repeated runs. Dashed lines denote perfect calibration.

tools for reasoning the uncertainty (Bishop, 2006; Kingma and Welling, 2014). In general, predictive uncertainty can be split into two types: aleatoric and epistemic uncertainty (Kendall and Gal, 2017). Aleatoric uncertainty arises from the data directly; e.g. sensor noise or motion artifacts. In regression, it is derived from the conditional log-likelihood under the maximum likelihood framework and can be captured by a deep model directly (see § 2.1). Epistemic uncertainty is caused by uncertainty in the model parameters due to a limited amount of training data (Bishop, 2006). A well-accepted approach to quantify epistemic uncertainty is variational inference with Monte Carlo (MC) dropout, where dropout is used at test time to sample from the approximate posterior (Gal and Ghahramani, 2016).

However, uncertainty obtained by deep BNNs tends to be miscalibrated, i.e. it does not correspond well with the model error (Laves et al., 2019). Fig. 1 shows calibration plots (predictive error vs. uncertainty) for uncalibrated and calibrated uncertainty estimates. The predictive uncertainty (taking into account both epistemic and aleatoric uncertainty) is underestimated and does not allow robust detection of uncertain predictions at test time.

Calibration of uncertainty in regression has been addressed in prior work. In (Kuleshov et al., 2018), inaccurate uncertainties from Bayesian models for regression are recalibrated using a technique inspired by Platt scaling. Given a pre-trained, miscalibrated model $\boldsymbol{H}$, an auxiliary model $\boldsymbol{R} : [0, 1]^d \to [0, 1]^d$ is trained, that yields a calibrated regressor $\boldsymbol{R} \circ \boldsymbol{H}$. In (Phan et al., 2018), this method was applied to bounding box regression. However, an auxiliary model with enough capacity will always be able to recalibrate, even if the predicted uncertainty is completely uncorrelated with the real uncertainty. Furthermore, Kuleshov et al. state that calibration via $\boldsymbol{R}$ is possible if enough independent and identically distributed (i.i.d.) data is available. In medical imaging, large data sets are usually hard to obtain, which can cause $\boldsymbol{R}$ to overfit the calibration set (as we will show later). This downside was addressed in (Levi et al., 2019), which is most related to our work. They proposed to scale the standard deviation of a Gaussian model to recalibrate aleatoric uncertainty. In contrast

to our work, they do not take into account epistemic uncertainty, which is an important source of uncertainty, especially when dealing with small data sets in medical imaging.

To the best of our knowledge, calibration of predictive uncertainty for regression tasks in medical imaging has not been addressed. Our main contributions are: (1) We analyze and provide theoretical background why deep models for regression are miscalibrated with regard to predictive uncertainty, (2) we suggest to use $\sigma$ *scaling* in a separate calibration phase to tackle underestimation of uncertainty, and (3) we perform extensive experiments on four different data sets to show the effectiveness of the proposed method.

## 2. Methods

In this section, we discuss estimation of aleatoric and epistemic uncertainty for regression and show, why uncertainty is systematically miscalibrated. We propose to use $\sigma$ scaling to jointly calibrate aleatoric and epistemic uncertainty.

### 2.1. Conditional Log-Likelihood for Regression

We revisit regression under the maximum posterior (MAP) framework to derive direct estimation of heteroscedastic aleatoric uncertainty. The goal of our regression model is to predict a target value $\boldsymbol{y}$ given some new input $\boldsymbol{x}$ and a training set $\mathcal{D}$ of $m$ inputs $\{\boldsymbol{x}_1, \ldots, \boldsymbol{x}_m\}$ and their corresponding (observed) target values $\{\boldsymbol{y}_1, \ldots, \boldsymbol{y}_m\}$. We assume that $\boldsymbol{y}$ has a Gaussian distribution $\mathcal{N}\left(\boldsymbol{y}; \hat{\boldsymbol{y}}(\boldsymbol{x}), \hat{\sigma}^2(\boldsymbol{x})\right)$ with mean equal to $\hat{\boldsymbol{y}}(\boldsymbol{x})$ and variance $\hat{\sigma}^2(\boldsymbol{x})$. A neural network with parameters $\boldsymbol{\theta}$

$$\boldsymbol{f_\theta}\left(\boldsymbol{x}\right) = \left[\hat{\boldsymbol{y}}(\boldsymbol{x}), \hat{\sigma}^2(\boldsymbol{x})\right], \ \hat{\boldsymbol{y}} \in \mathbb{R}^d, \ \hat{\sigma}^2 \geq 0 \tag{1}$$

outputs these values for a given input. By assuming a Gaussian prior over the parameters $\boldsymbol{\theta} \sim \mathcal{N}(\boldsymbol{\theta}; \boldsymbol{0}, \lambda^{-1}\boldsymbol{I})$, MAP estimation becomes maximum-likelihood estimation with added weight decay (Bishop, 2006). With $m$ i.i.d. random samples, the conditional log-likelihood is given by

$$\sum_{i=1}^{m} \log\left(\frac{1}{\sqrt{2\pi}\hat{\sigma}_{\boldsymbol{\theta}}^{(i)}} \exp\left\{-\frac{\left\|\boldsymbol{y}^{(i)} - \hat{\boldsymbol{y}}_{\boldsymbol{\theta}}^{(i)}\right\|^2}{2\left(\hat{\sigma}_{\boldsymbol{\theta}}^{(i)}\right)^2}\right\}\right) \tag{2}$$

$$= -\frac{m}{2}\log\left(2\pi\right) - \sum_{i=1}^{m}\log\left(\hat{\sigma}_{\boldsymbol{\theta}}^{(i)}\right) + \frac{1}{2\left(\hat{\sigma}_{\boldsymbol{\theta}}^{(i)}\right)^2}\left\|\boldsymbol{y}^{(i)} - \hat{\boldsymbol{y}}_{\boldsymbol{\theta}}^{(i)}\right\|^2 . \tag{3}$$

The dependence on $\boldsymbol{x}$ has been omitted to simplify the notation. Maximizing the log-likelihood in Eq. (3) w.r.t. $\boldsymbol{\theta}$ is equivalent to minimizing the negative log-likelihood (NLL), which leads to the following optimization criterion (with weight decay)

$$\mathcal{L}_{\mathrm{G}}(\boldsymbol{\theta}) = \sum_{i=1}^{m}\left(\hat{\sigma}_{\boldsymbol{\theta}}^{(i)}\right)^{-2}\left\|\boldsymbol{y}^{(i)} - \hat{\boldsymbol{y}}_{\boldsymbol{\theta}}^{(i)}\right\|^2 + \log\left(\left(\hat{\sigma}_{\boldsymbol{\theta}}^{(i)}\right)^2\right) . \tag{4}$$

Here, $\hat{\boldsymbol{y}}_{\boldsymbol{\theta}}$ and $\hat{\sigma}_{\boldsymbol{\theta}}$ are estimated jointly by finding $\boldsymbol{\theta}$ that minimizes Eq. (4). This can be achieved using gradient descent in a standard training procedure. In this case, $\hat{\sigma}_{\boldsymbol{\theta}}$ captures the uncertainty that is inherent in the data (aleatoric uncertainty). To avoid numerical instability due to potential division by zero, we directly estimate $\log\hat{\sigma}^2(\boldsymbol{x})$ and implement Eq. (4) in similar practice to Kendall and Gal (2017).

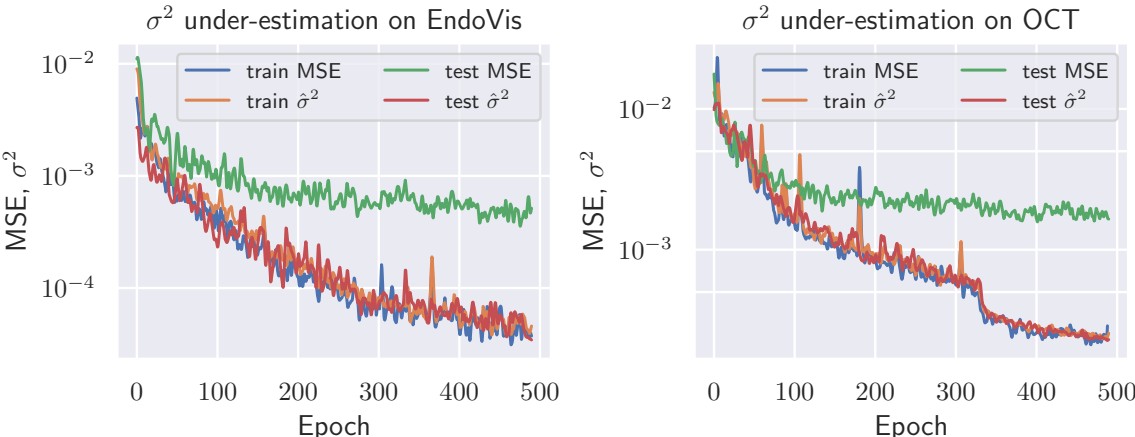

Figure 2: Biased estimation of aleatoric uncertainty $\sigma^2$. The deep model overfits estimation of $\boldsymbol{y}$ on the training set. On unseen test data, the MSE of predictive mean is higher and $\sigma^2$ is underestimated. Early stopping (e.g. at epoch 50) would result in an unbiased estimator, but this would not be optimal in terms of test MSE.

## 2.2. Biased estimation of $\sigma$

Ignoring the dependence through $\boldsymbol{\theta}$, the solution to Eq. (4) decouples estimation of $\hat{\boldsymbol{y}}$ and $\hat{\sigma}$. In case of a Gaussian likelihood, minimizing (4) w.r.t. $\hat{\boldsymbol{y}}^{(i)}$ yields

$$\hat{\boldsymbol{y}}^{(i)} = \underset{\hat{\boldsymbol{y}}^{(i)}}{\arg\min} \, \mathcal{L}_{\mathrm{G}} = \boldsymbol{y}^{(i)} \ \forall i \ . \tag{5}$$

Minimizing (4) w.r.t. $(\hat{\sigma}^{(i)})^2$ yields

$$\left(\hat{\sigma}^{(i)}\right)^2 = \underset{(\hat{\sigma}^{(i)})^2}{\arg\min} \, \mathcal{L}_{\mathrm{G}} = \|\boldsymbol{y}^{(i)} - \hat{\boldsymbol{y}}^{(i)}\|^2 \ \forall i \ . \tag{6}$$

That is, estimation of $\sigma^2$ should perfectly reflect the squared error. However, in (6) $\sigma^2$ is estimated relative to the estimated mean $\hat{\boldsymbol{y}}$ and therefore biased. In fact, the maximum likelihood solution systematically underestimates $\sigma^2$, which is a phenomenon of overfitting the training set (Bishop, 2006). The squared error $\|\boldsymbol{y} - \hat{\boldsymbol{y}}\|^2$ will be lower on the training set and $\hat{\sigma}^2$ on new samples will be systematically too low (see Fig. 2). This is a problem especially in deep learning, where large models have millions of parameters and tend to overfit. Bias in estimation of $\sigma^2$ can be corrected by a scaling factor. Rescaling for unbiased estimation of the population variance via computing the sample variance is famously known as *Bessel's correction*. In our case, we introduce a simple learnable scalar parameter to rescale the biased estimation of $\sigma^2$.

### 2.3. $\sigma$ Scaling for Aleatoric Uncertainty

We first derive $\sigma$ scaling for aleatoric uncertainty. Using a Gaussian model, we scale the standard deviation $\sigma$ with a scalar value $s$ to recalibrate the probability density function

$$p\left(\boldsymbol{y}|\boldsymbol{x}; \hat{\boldsymbol{y}}(\boldsymbol{x}), \hat{\sigma}^2(\boldsymbol{x})\right) = \mathcal{N}\left(\boldsymbol{y}; \hat{\boldsymbol{y}}(\boldsymbol{x}), (s \cdot \hat{\sigma}(\boldsymbol{x}))^2\right) \ . \tag{7}$$

Now, the conditional log-likelihood is given by

$$\sum_{i=1}^{m} \log p\left(\boldsymbol{y}^{(i)}|\boldsymbol{x}; \hat{\boldsymbol{y}}_{\boldsymbol{\theta}}^{(i)}, \left(s \cdot \hat{\sigma}_{\boldsymbol{\theta}}^{(i)}\right)^2\right) \ . \tag{8}$$

This results in the following optimization objective

$$\mathcal{L}_{\mathrm{G}}(s) = m \log(s) + \tfrac{1}{2} s^{-2} \sum_{i=1}^{m} \left(\hat{\sigma}_{\boldsymbol{\theta}}^{(i)}\right)^{-2} \big\| \boldsymbol{y}^{(i)} - \hat{\boldsymbol{y}}_{\boldsymbol{\theta}}^{(i)} \big\|^2 \ . \tag{9}$$

Eq. (9) is optimized w.r.t. $s$ with fixed $\boldsymbol{\theta}$ using gradient descent in a separate calibration phase after training to calibrate aleatoric uncertainty measured by $\hat{\sigma}_{\boldsymbol{\theta}}^2$. In case of a single scalar, the solution to Eq. (9) can also be written in closed form as

$$s = \pm \sqrt{\frac{1}{m} \sum_{i=1}^{m} \left(\hat{\sigma}_{\boldsymbol{\theta}}^{(i)}\right)^{-2} \big\| \boldsymbol{y}^{(i)} - \hat{\boldsymbol{y}}_{\boldsymbol{\theta}}^{(i)} \big\|^2} \ . \tag{10}$$

We apply $\sigma$ scaling to jointly calibrate aleatoric and epistemic uncertainty in the next section.

### 2.4. Well-Calibrated Estimation of Predictive Uncertainty

So far we have assumed a MAP point estimate for $\boldsymbol{\theta}$ which does not consider uncertainty in the parameters. To quantify both aleatoric and epistemic uncertainty, we extend $\boldsymbol{f}_{\boldsymbol{\theta}}$ into a fully Bayesian model under the variational inference framework with Monte Carlo dropout (Gal and Ghahramani, 2016). In MC dropout, the model $\boldsymbol{f}_{\tilde{\boldsymbol{\theta}}}$ is trained with dropout (Srivastava et al., 2014) and dropout is applied at test time by performing $N$ stochastic forward passes to sample from the approximate Bayesian posterior $\tilde{\boldsymbol{\theta}} \sim q(\boldsymbol{\theta})$. Following (Kendall and Gal, 2017), we use MC integration to approximate the predictive variance

$$\hat{\Sigma}^2 = \underbrace{\frac{1}{N} \sum_{n=1}^{N} \left(\hat{\boldsymbol{y}}_n - \frac{1}{N} \sum_{n=1}^{N} \hat{\boldsymbol{y}}_n\right)^2}_{\text{epistemic}} + \underbrace{\frac{1}{N} \sum_{n=1}^{N} \hat{\sigma}_n^2}_{\text{aleatoric}} \tag{11}$$

and use $\hat{\Sigma}^2$ as a measure of predictive uncertainty. If the neural network has multiple outputs $(d > 1)$, the predictive variance is calculated per output and the mean across $d$ forms the final uncertainty value. We expect $\hat{\Sigma}^2$ to reflect the squared error of $\hat{\boldsymbol{y}}$ and define perfect calibration of predictive uncertainty as

$$\mathbb{E}_{\boldsymbol{x}, \boldsymbol{y}} \left[ \|\boldsymbol{y} - \hat{\boldsymbol{y}}\|^2 \,\big|\, \hat{\Sigma}^2 = \Sigma^2 \right] = \Sigma^2 \quad \forall \left\{ \Sigma^2 \in \mathbb{R} \,|\, \Sigma^2 \geq 0 \right\} \ , \tag{12}$$

which extends the definition by (Levi et al., 2019). For example, in a batch of images all predicted with $\hat{\Sigma}^2 = 0.5$, the expectation of the squared error should equal 0.5. Eq. (11) is an unbiased estimator of the approximate predictive variance (see proof in Appendix C). However, even in deep learning with Bayesian principles, the approximate posterior predictive distribution can overfit on small data sets. In practice, this results in underestimation of the predictive uncertainty.

One could regularize overfitting by early stopping at minimal loss (Eq. (4)) on the validation set, which would circumvent underestimation of $\sigma^2$. However, our experiments show that early stopping is not optimal with regard to the squared error of $\hat{\boldsymbol{y}}$ on both training and testing data (see Fig. 2). In contrast, the model with lowest mean error on the validation set underestimates predictive uncertainty considerably. Therefore, we apply $\sigma$ scaling to recalibrate the predictive uncertainty $\hat{\Sigma}^2$. This allows a lower squared error but reduces underestimation of uncertainty as shown experimentally in the following section.

### 2.5. Expected Uncertainty Calibration Error for Regression

We extend the definition of miscalibrated uncertainty for classification (Laves et al., 2019) to quantify miscalibration of uncertainty in regression

$$\mathbb{E}_{\hat{\Sigma}^2}\left[\left|\left(\|\boldsymbol{y} - \hat{\boldsymbol{y}}\|^2 \,\big|\, \hat{\Sigma}^2 = \Sigma^2\right) - \Sigma^2\right|\right] \quad \forall \left\{\Sigma^2 \in \mathbb{R} \,\big|\, \Sigma^2 \geq 0\right\} \; . \tag{13}$$

On finite data sets, this can be approximated with the expected uncertainty calibration error (UCE) for regression. Following (Guo et al., 2017), the uncertainty output $\hat{\Sigma}^2$ of a deep model is partitioned into $M$ bins with equal width. A weighted average of the difference between the predictive error and uncertainty is used:

$$\text{UCE} := \sum_{m=1}^{M} \frac{|B_m|}{n} \big|\text{err}(B_m) - \text{uncert}(B_m)\big| \; , \tag{14}$$

with number of inputs $n$ and set of indices $B_m$ of inputs, for which the uncertainty falls into the bin. The error per bin is defined as

$$\text{err}(B_m) := \frac{1}{|B_m|} \sum_{i \in B_m} \left\|\boldsymbol{y}_i - \hat{\boldsymbol{y}}_i\right\|^2 \; , \tag{15}$$

and the uncertainty per bin is defined as

$$\text{uncert}(B_m) := \frac{1}{|B_m|} \sum_{i \in B_m} \hat{\Sigma}_i^2 \; . \tag{16}$$

Throughout this work, UCE is given in %. Additionally, we plot $\text{err}(B_m)$ vs. $\text{uncert}(B_m)$ to create calibration diagrams.

## 3. Experiments & Results

We use four data sets and three different deep network architectures to evaluate recalibration with $\sigma$ scaling. The last linear layer of all networks is replaced by two linear layers predicting

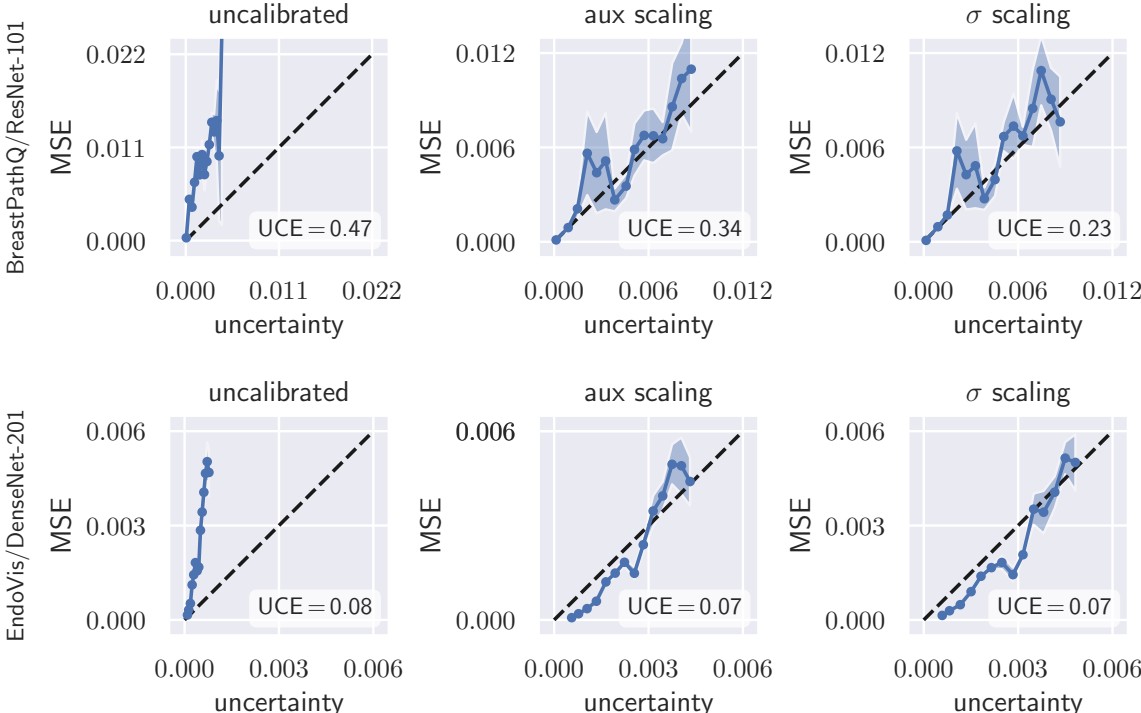

Figure 3: Calibration plots for ResNet-101 on BreastPathQ (top row) and DenseNet-201 on EndoVis (bottom row). Aux scaling tends to overfit the calibration set, which results in higher UCE compared to simple $\sigma$ scaling. Dashed lines denote perfect calibration.

$\hat{\boldsymbol{y}}$ and $\hat{\sigma}^2$ as described in § 2.1. For MC dropout, we use dropout before the last linear layers and at different locations, depending on the architecture. Dropout is additionally added after each of the four layers of stacked residual blocks in ResNet (He et al., 2016). In DenseNet and EfficientNet, we use the default configuration of dropout during training and testing (Huang et al., 2017; Tan and Le, 2019). The networks are trained until no further decrease in mean squared error (MSE) on the validation set can be observed.

The data sets were selected to represent various regression tasks in medical imaging with different dimension $d$ of target value $\boldsymbol{y} \in \mathbb{R}^d$: (1) tumor cellularity in H & E stained whole slides of cancerous breast tissue from BreastPathQ SPIE challenge data set ($d = 1$) (Martel et al., 2019), (2) hand CT age regression from the RSNA pediatric bone age data set ($d = 1$) (Halabi et al., 2019), (3) surgical instrument tracking on endoscopic images from EndoVis endoscopic vision challenge 2015 data set ($d = 2$), and (4) 6DoF needle pose estimation on optical coherence tomography (OCT) scans from our own data set[1] ($d = 6$). All outputs are normalized such that $\boldsymbol{y} \in [0, 1]^d$. More details on the training procedure can be found in Appendix D.

---

1. Our OCT pose estimation data set is publicly available at github.com/mlaves/3doct-pose-dataset

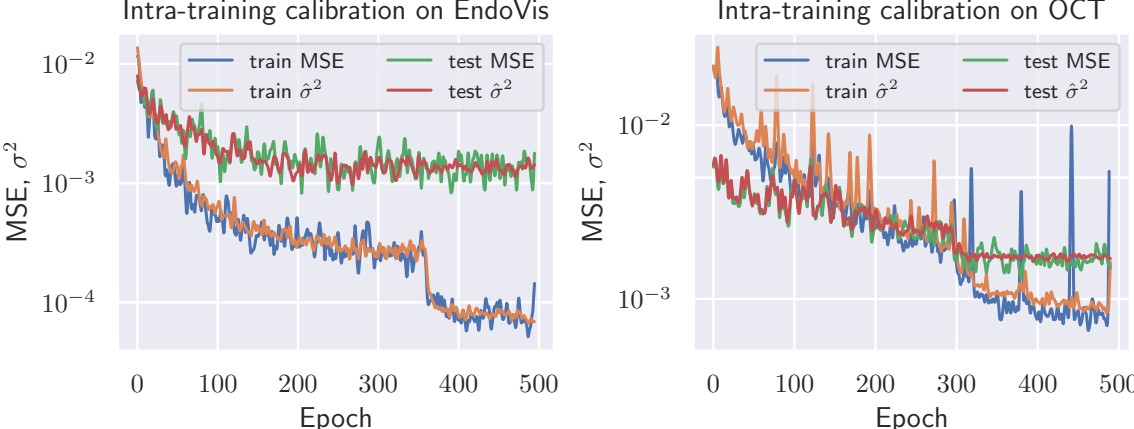

Figure 4: (Left) Intra-training calibration of aleatoric uncertainty with $\sigma$ scaling. The deep model no longer underestimates $\hat{\sigma}^2$ on unseen test data. (Right) The MSE of predictive mean is higher and $\sigma^2$ is underestimated. Note: Calibration is only applied at test time.

Table 1: Values of negative log-likelihood of uncalibrated and calibrated models. If the uncalibrated model already achieves low NLL, aux scaling may overfit the calibration set, resulting in worse NLL on the test set.

|  |  | uncalibrated | aux scaling | $\sigma$ scaling |
|---|---|---|---|---|
| ResNet-101/ | calibration set NLL | -2.26 | **-4.92** | -4.88 |
| BreastPathQ | test set NLL | -2.82 | -4.89 | **-5.02** |
| EfficientNet-B4/ | calibration set NLL | -4.92 | **-5.91** | -5.88 |
| EndoVis | test set NLL | -5.93 | -6.17 | **-6.24** |

Calibration is performed after training in a separate calibration phase using the validation data set. We plug the predictive uncertainty $\hat{\Sigma}^2$ into Eq. (9) and minimize w.r.t. $s$. Additionally, we compare $\sigma$ scaling to a more powerful auxiliary recalibration model $\boldsymbol{R}$ consisting of a two-layer fully-connected network with 16 hidden units and ReLU activations (inspired by (Kuleshov et al., 2018), see §1).

To quantify miscalibration, we use the proposed expected uncertainty calibration error for regression. We visualize (mis-)calibration in Fig. 1 and Fig. 3 using calibration diagrams, which show predictive uncertainty vs. predictive error (MSE). The discrepancy to the identity function reveals miscalibration. Fig. 4 uses intra-training calibration of aleatoric uncertainty to show closing the gap between test MSE and uncertainty. Tab. 2 reports negative log-likelihood and UCE values of all data set/model combinations on the respective test sets. Fig. 5 shows a practical example from the EndoVis test set. Figures for all configurations are listed in Appendix F.

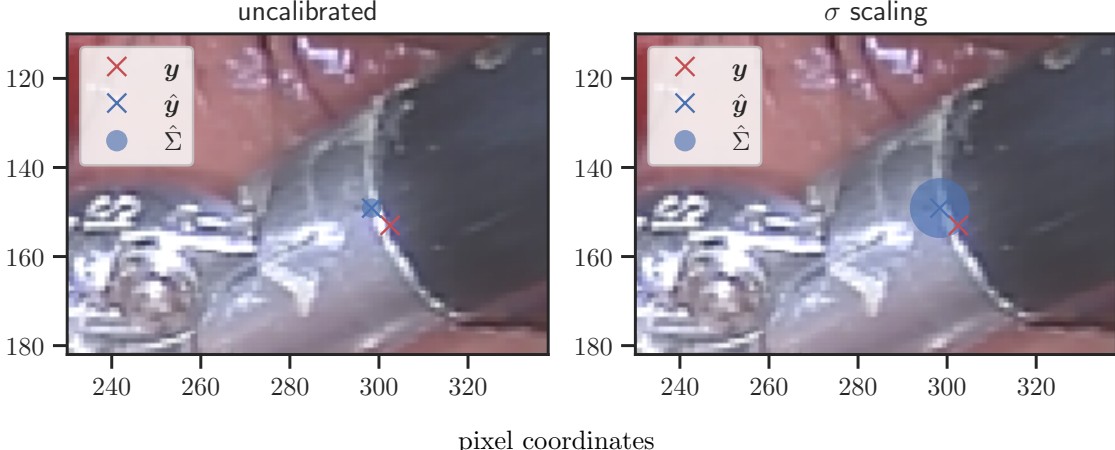

Figure 5: Example result from EndoVis test set. The task is to predict pixel coordinates of the forceps shaft center. Before calibration, the uncertainty is underestimated and the true instrument position $y$ does not fall into the predictive uncertainty region around $\hat{y}$. After calibration with $\sigma$ scaling, the uncertainty better reflects the predictive error.

Table 2: Test set results for different datasets and model architectures (averaged over 5 runs). The values of the negative log-likelihood and uncertainty calibration error quantify miscalibration. In addition, the resulting $s$ for $\sigma$ scaling is given.

| | | | uncalibrated | | aux scaling | | $\sigma$ scaling | | |
|---|---|---|---|---|---|---|---|---|---|
| Data Set | Model | MSE | NLL | UCE | NLL | UCE | NLL | UCE | s |
| | ResNet-101 | 6.0e-3 | -2.90 | 0.47 | -5.17 | 0.34 | -5.16 | **0.23** | 2.39 |
| BreastPathQ | DenseNet-201 | 6.2e-3 | -5.66 | 0.28 | -6.04 | 0.42 | -5.77 | **0.15** | 1.31 |
| | EfficientNet-B4 | 5.9e-3 | -4.75 | 0.44 | -6.38 | 0.13 | -5.62 | **0.11** | 1.79 |
| | ResNet-101 | 5.1e-3 | -3.99 | 0.28 | -4.34 | **0.06** | -4.34 | **0.06** | 1.40 |
| BoneAge | DenseNet-201 | 3.5e-3 | -0.84 | 0.29 | -4.71 | **0.04** | -4.71 | **0.04** | 2.53 |
| | EfficientNet-B4 | 3.5e-3 | 6.34 | 0.32 | -4.75 | **0.06** | -4.64 | 0.18 | 3.91 |
| | ResNet-101 | 4.0e-4 | -3.85 | **0.03** | -6.79 | 0.04 | -6.73 | 0.04 | 3.46 |
| EndoVis | DenseNet-201 | 1.1e-3 | -4.97 | 0.08 | -6.01 | **0.07** | -6.04 | **0.07** | 2.58 |
| | EfficientNet-B4 | 8.9e-4 | -5.94 | 0.05 | -6.18 | **0.04** | -6.17 | **0.04** | 1.78 |
| | ResNet-101 | 2.0e-3 | -3.38 | 0.15 | -5.24 | **0.02** | -5.24 | **0.02** | 2.13 |
| OCT | DenseNet-201 | 1.3e-3 | -5.51 | 0.04 | -5.61 | **0.01** | -5.61 | 0.02 | 1.27 |
| | EfficientNet-B4 | 1.4e-3 | -4.25 | 0.10 | -5.58 | **0.01** | -5.57 | **0.01** | 1.93 |

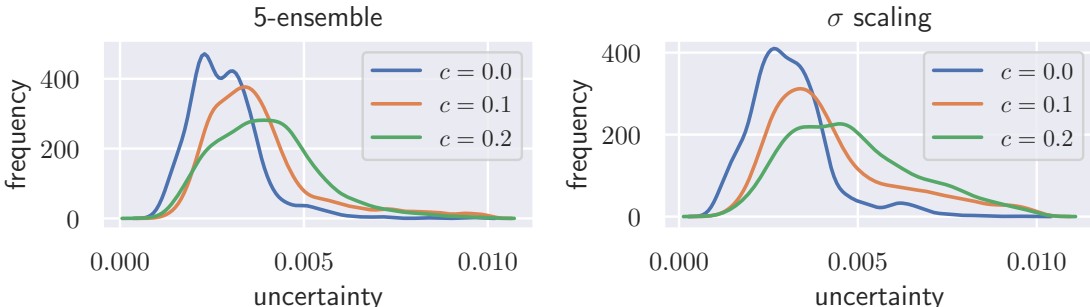

Figure 6: Histograms of the uncertainties for out-of-distribution detection with DenseNet-201 on BoneAge test set. (Left) Uncertainties from a non-Bayesian ensemble of five DenseNets and (right) Bayesian uncertainties calibrated with $\sigma$ scaling. The distribution shift has been created by additive Gaussian noise with $\mathcal{N}(c, c^2)$.

### 3.1. Detection of Out-of-Distribution Data and Unreliable Predictions

Deep neural networks only yield reliable predictions for data which follow the same distribution as the training data. A shift in distribution could occur when a model trained on CT data from a specific CT device is applied to data from another manufacturer's CT device, for example. To create a distribution shift, we add Gaussian noise with $\mathcal{N}(c, c^2)$ to the BoneAge data and report histograms of the uncertainties for $c \in \{0.0, 0.1, 0.2\}$ (see Fig. 6). Lakshminarayanan et al. (2017) state that deep ensembles provide better-calibrated uncertainty than Bayesian neural networks with MC dropout variational inference. We therefore train an ensemble of 5 randomly initialized DenseNet-201 and compare Bayesian uncertainty with $\sigma$ scaling to ensemble uncertainty under distribution shift.

Additionally, we apply the well-calibrated models to detect and reject uncertain predictions, as crucial decisions in medical practice should only be made on the basis of reliable predictions. An uncertainty threshold $\Sigma_{\max}^2$ is defined and all predictions from the test set are rejected where $\hat{\Sigma}^2 > \Sigma_{\max}^2$ (see Fig. 7). From this, a decrease in overall MSE is expected. We additionally compare rejection on the basis of $\sigma$ scaled uncertainty to uncertainty from the aforementioned ensemble.

### 4. Discussion & Conclusion

In this paper, well-calibrated predictive uncertainty in medical imaging obtained by variational inference with deep Bayesian models is discussed. Both calibration methods considerably reduce miscalibration of predictive uncertainty in terms of UCE and NLL. If the model is already well-calibrated (see BreastPathQ/DenseNet-201 in Tab. 2), aux scaling can slightly increase UCE. In such cases, we often observe the more powerful auxiliary model $\boldsymbol{R}$ to overfit the calibration set (see Tab. 1). This results in aux scaling yielding the lowest NLL on the calibration set, which is however outperformed on the test set by $\sigma$ scaling, or even by the uncalibrated model. If the deep model is already well-calibrated, $\sigma$ scaling does not negatively affect the calibration, which results in $s \to 1$.

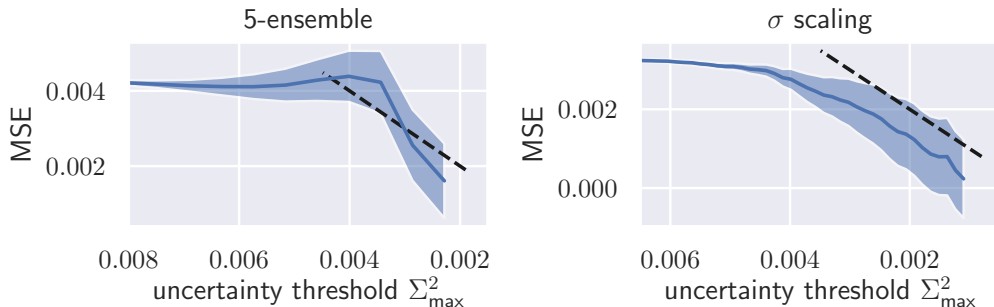

Figure 7: Rejection of uncertain predictions with DenseNet-201 on BoneAge test set with $\hat{\Sigma}^2 > \Sigma^2_{\mathrm{max}}$. The shaded area width visualizes the percentage of rejected samples. The dashed line visualizes linear relationship.

Well-calibrated uncertainty from MC dropout is able to reliably detect a shift in the data distribution. The results are comparable to those from a deep ensemble, but without the need to train multiple individual models on the same data set. This is in contrast to what was reported by Lakshminarayanan et al. (2017). BNNs calibrated with $\sigma$ scaling even outperform ensembles in the rejection task (see Fig. 7). In case of $\sigma$ scaling, the test set MSE decreases monotonically as a function of the uncertainty threshold, whereas the ensemble initially shows an increasing MSE.

$\sigma$ scaling is simple to implement, does not change the predictive mean $\hat{\boldsymbol{y}}$, and does not affect the model accuracy. It is preferable to regularization (e.g. early stopping) or more complex recalibration methods in calibrated uncertainty estimation with Bayesian deep learning. The disconnection between test MSE and test NLL can successfully be closed, which creates highly accurate models with reliable uncertainty estimates.

Predictive uncertainty should be considered in any medical imaging task that is approached with deep learning. Well-calibrated uncertainty is of great importance for decision-making and is anticipated to increase patient safety. It allows to robustly reject unreliable predictions or out-of-distribution samples. However, there are many factors (e.g. network capacity, weight decay, dropout configuration) influencing the uncertainty that have not been discussed here and will be addressed in future work.

## Acknowledgments

We thank Vincent Modes for his insightful comments. This research has received funding from the European Union as being part of the ERDF OPhonLas project.

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

## Appendix A. Laplacian Model

Using $\mathsf{Laplace}(\hat{\boldsymbol{y}}(\boldsymbol{x}), \hat{\sigma}(\boldsymbol{x}))$ as model, the conditional log-likelihood is given by

$$\sum_{i=1}^{m} \log \left( \frac{1}{2\hat{\sigma}_{\boldsymbol{\theta}}^{(i)}} \exp \left\{ -\frac{\left|\boldsymbol{y}^{(i)} - \hat{\boldsymbol{y}}_{\boldsymbol{\theta}}^{(i)}\right|}{\hat{\sigma}_{\boldsymbol{\theta}}^{(i)}} \right\} \right) \tag{17}$$

$$\tag{18}$$

which results in the following optimization criterion

$$\mathcal{L}_{\mathrm{L}}(\boldsymbol{\theta}) = \sum_{i=1}^{m} \frac{1}{\hat{\sigma}_{\boldsymbol{\theta}}^{(i)}} \left|\boldsymbol{y}^{(i)} - \hat{\boldsymbol{y}}_{\boldsymbol{\theta}}^{(i)}\right| + \log\left(\hat{\sigma}_{\boldsymbol{\theta}}^{(i)}\right) . \tag{19}$$

Using $\mathcal{L}_{\mathrm{L}}(\boldsymbol{\theta})$ instead of $\mathcal{L}_{\mathrm{G}}(\boldsymbol{\theta})$ results in performing an L1 metric on the predictive mean. In some cases, this led to better results. However, we have not conducted extensive experiments with it and leave it to future work.

## Appendix B. Derivation of $\sigma$ Scaling

See § 2.3. Using a Gaussian model, we scale the standard deviation $\sigma$ with a scalar value $s$ to calibrate the PDF

$$p\left(\boldsymbol{y}|\boldsymbol{x}; \hat{\boldsymbol{y}}(x), \hat{\sigma}^2(x)\right) = \mathcal{N}\left(\boldsymbol{y}; \hat{\boldsymbol{y}}(x), (s \cdot \hat{\sigma}(x))^2\right) . \tag{20}$$

The conditional log-likelihood is given by

$$\sum_{i=1}^{m} \log \left( \frac{1}{\sqrt{2\pi} s\hat{\sigma}_{\boldsymbol{\theta}}^{(i)}} \exp \left( \frac{\left\|\boldsymbol{y}^{(i)} - \hat{\boldsymbol{y}}_{\boldsymbol{\theta}}^{(i)}\right\|^2}{2\left(s\hat{\sigma}_{\boldsymbol{\theta}}^{(i)}\right)^2} \right) \right) \tag{21}$$

$$= -\frac{m}{2}\log(2\pi) - \sum_{i=1}^{m} \log\left(s\hat{\sigma}_{\boldsymbol{\theta}}^{(i)}\right) + \frac{1}{2}\left(s\hat{\sigma}_{\boldsymbol{\theta}}^{(i)}\right)^{-2} \cdot \left\|\boldsymbol{y}^{(i)} - \hat{\boldsymbol{y}}_{\boldsymbol{\theta}}^{(i)}\right\|^2 \tag{22}$$

This results in the following optimization objective (ignoring constants):

$$\mathcal{L}_{\mathrm{G}}(s) = m\log(s) + \tfrac{1}{2}s^{-2}\sum_{i=1}^{m}(\hat{\sigma}_{\boldsymbol{\theta}}^{(i)})^{-2}\left\|\boldsymbol{y}^{(i)} - \hat{\boldsymbol{y}}_{\boldsymbol{\theta}}^{(i)}\right\|^2 . \tag{23}$$

Using a Laplacian model, the optimization criterion follows as

$$\mathcal{L}_{\mathrm{L}}(s) = m\log(s) + s^{-1}\sum_{i=1}^{m} \frac{1}{\hat{\sigma}_{\boldsymbol{\theta}}^{(i)}}\left|\boldsymbol{y}^{(i)} - \hat{\boldsymbol{y}}_{\boldsymbol{\theta}}^{(i)}\right| . \tag{24}$$

Eq. (23) and (24) are optimized w.r.t. $s$ with fixed $\boldsymbol{\theta}$ using gradient descent in a separate calibration phase after training. The solution to Eq. (23) can also be written in closed form as

$$s_{\mathrm{G}} = \pm\sqrt{\frac{1}{m}\sum_{i=1}^{m}\left(\hat{\sigma}_{\boldsymbol{\theta}}^{(i)}\right)^{-2}\left\|\boldsymbol{y}^{(i)} - \hat{\boldsymbol{y}}_{\boldsymbol{\theta}}^{(i)}\right\|^2} \tag{25}$$

and the solution to Eq. (24) follows as

$$s_{\mathrm{L}} = \frac{1}{m} \sum_{i=1}^{m} \frac{1}{\hat{\sigma}_{\boldsymbol{\theta}}^{(i)}} \left| \boldsymbol{y}^{(i)} - \hat{\boldsymbol{y}}_{\boldsymbol{\theta}}^{(i)} \right| \; , \tag{26}$$

respectively. We apply $\sigma$ scaling to jointly calibrate aleatoric and epistemic uncertainty as described in § 2.4.

## Appendix C. Unbiased Estimator of the Approximate Predictive Variance

We show that the expectation of the predictive sample variance from MC dropout, as given in (Kendall and Gal, 2017), equals the true variance of the approximate posterior predictive distribution.

**Proposition 1** *Given $N$ MC dropout samples $\boldsymbol{f}_{\boldsymbol{\theta}_n} = [\hat{\boldsymbol{y}}_n, \hat{\sigma}_n^2]$ from our approximate predictive distribution $p(\boldsymbol{y}^*|\boldsymbol{x}^*, \mathcal{D}) = \mathcal{N}(\boldsymbol{y}^*; \boldsymbol{y}, \Sigma^2)$, the predictive sample variance*

$$\hat{\Sigma}^2 = \frac{1}{N} \sum_{n=1}^{N} \left( \hat{\boldsymbol{y}}_n - \frac{1}{N} \sum_{n=1}^{N} \hat{\boldsymbol{y}}_n \right)^2 + \frac{1}{N} \sum_{n=1}^{N} \hat{\sigma}_n^2 \tag{27}$$

*is an unbiased estimator of the approximate predictive variance.*

**Proof**

$$\mathbb{E}\left[\hat{\Sigma}^2\right] = \mathbb{E}\left[ \frac{1}{N} \sum_{n=1}^{N} \left( \hat{\boldsymbol{y}}_n - \frac{1}{N} \sum_{n=1}^{N} \hat{\boldsymbol{y}}_n \right)^2 + \frac{1}{N} \sum_{n=1}^{N} \hat{\sigma}_n^2 \right] \tag{28}$$

$$= \mathbb{E}\left[ \frac{1}{N} \sum_{n=1}^{N} \left( \hat{\boldsymbol{y}}_n - \frac{1}{N} \sum_{n=1}^{N} \hat{\boldsymbol{y}}_n \right)^2 \right] + \mathbb{E}\left[ \frac{1}{N} \sum_{n=1}^{N} \hat{\sigma}_n^2 \right] \tag{29}$$

$$\text{with} \quad \frac{1}{N} \sum_{n=1}^{N} \hat{\boldsymbol{y}}_n = \bar{\boldsymbol{y}} \quad \text{follows} \tag{30}$$

$$= \mathbb{E}\left[ \frac{1}{N} \sum_{n=1}^{N} (\hat{\boldsymbol{y}}_n - \bar{\boldsymbol{y}})^2 \right] + \hat{\sigma}^2 \tag{31}$$

$$= \mathbb{E}\left[ \frac{1}{N} \sum_{n=1}^{N} (\hat{\boldsymbol{y}}_n - \bar{\boldsymbol{y}})^2 + \bar{\boldsymbol{y}}^2 - \bar{\boldsymbol{y}}^2 + \boldsymbol{y}^2 - \boldsymbol{y}^2 + 2\bar{\boldsymbol{y}}\boldsymbol{y} - 2\bar{\boldsymbol{y}}\boldsymbol{y} \right] + \hat{\sigma}^2 \tag{32}$$

$$= \mathbb{E}\left[ \frac{1}{N} \sum_{n=1}^{N} (\hat{\boldsymbol{y}}_n - \boldsymbol{y})^2 - (\bar{\boldsymbol{y}} - \boldsymbol{y})^2 \right] + \hat{\sigma}^2 \tag{33}$$

$$= \mathbb{E}\left[ \frac{1}{N} \sum_{n=1}^{N} (\hat{\boldsymbol{y}}_n - \boldsymbol{y})^2 \right] - \mathbb{E}\left[ (\bar{\boldsymbol{y}} - \boldsymbol{y})^2 \right] + \hat{\sigma}^2 \tag{34}$$

$$= \Sigma^2 - \hat{\sigma}^2 + \hat{\sigma}^2 \tag{35}$$

$$\mathbb{E}\left[\hat{\Sigma}^2\right] = \Sigma^2 \tag{36}$$

Note that the predicted aleatoric uncertainty $\hat{\sigma}^2$ equals the expected squared error when trained $\boldsymbol{f}_\theta$ by minimizing NLL, thus $\mathbb{E}[(\bar{\boldsymbol{y}} - \boldsymbol{y})^2] = \hat{\sigma}^2$. ∎

## Appendix D. Training Procedure

The model implementations from PyTorch 1.3 (Paszke et al., 2019) are used and trained with the following settings:

- training for 500 epochs with batch size of 16
- Adam optimizer with initial learn rate of $3 \cdot 10^{-4}$ and weight decay with $\lambda = 10^{-7}$
- reduce-on-plateau learn rate scheduler (patience of 20 epochs) with factor of 0.1
- in MC dropout, $N = 25$ forward passes were performed with dropout with $p = 0.5$ used for ResNet (as described in (Gal and Ghahramani, 2016)). In DenseNet ($p = 0.2$) and EfficientNet ($p = 0.4$) standard dropout $p$ of the architecture is used.
- Additional validation and test sets are used if provided by the data sets; otherwise, a train/validation/test split of approx. $50\% / 25\% / 25\%$ is used

## Appendix E. 3D OCT Needle Pose Data Set

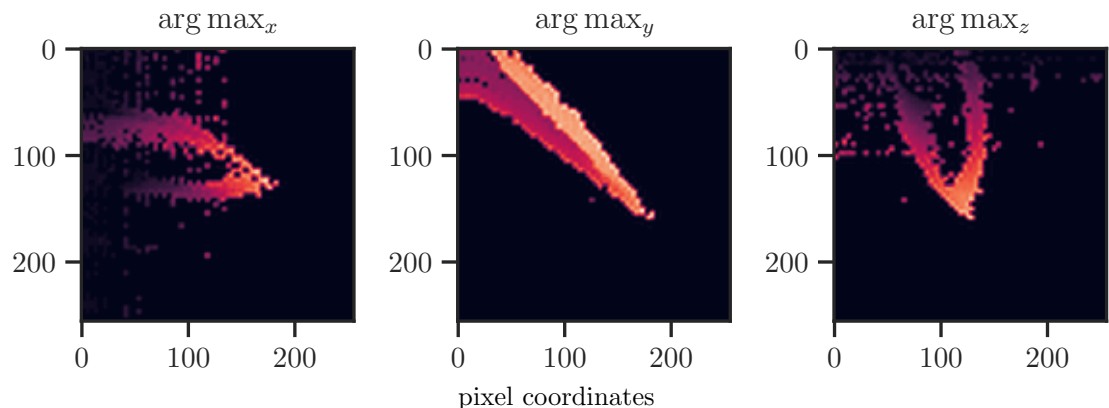

Figure 8: Example image from OCT data set showing arg max projections of a surgical needle tip acquired by optical coherence tomography.

Our data set was created by attaching a surgical needle to a high-precision six-axis hexapod robot (H-826, Physik Instrumente GmbH & Co. KG, Germany) and observing the needle tip with 3D optical coherence tomography (OCS1300SS, Thorlabs Inc., USA). The data set consists of 5,000 OCT acquisitions with $64 \times 64 \times 512$ voxels, covering a volume of approx. $3 \times 3 \times 3\,\text{mm}^3$. Each acquisition is taken at a different robot configuration and labeled with the corresponding 6DoF pose. To process the volumetric data with CNNs for planar images, we calculate 3 planar projections along the spatial dimensions using the arg max operator, scale them to equal size and stack them together as three-channel

image (see Fig. 8). A similar approach was presented in (Laves et al., 2017) and (Gessert et al., 2018). The data are characterized by a high amount of speckle noise, which is a typical phenomenon in optical coherence tomography. The data set is publicly available at github.com/mlaves/3doct-pose-dataset.

## Appendix F. Additional Calibration Diagrams

All test set runs have been repeated 5 times. Solid lines denote mean and shaded areas denote standard deviation calculated from the repeated runs.

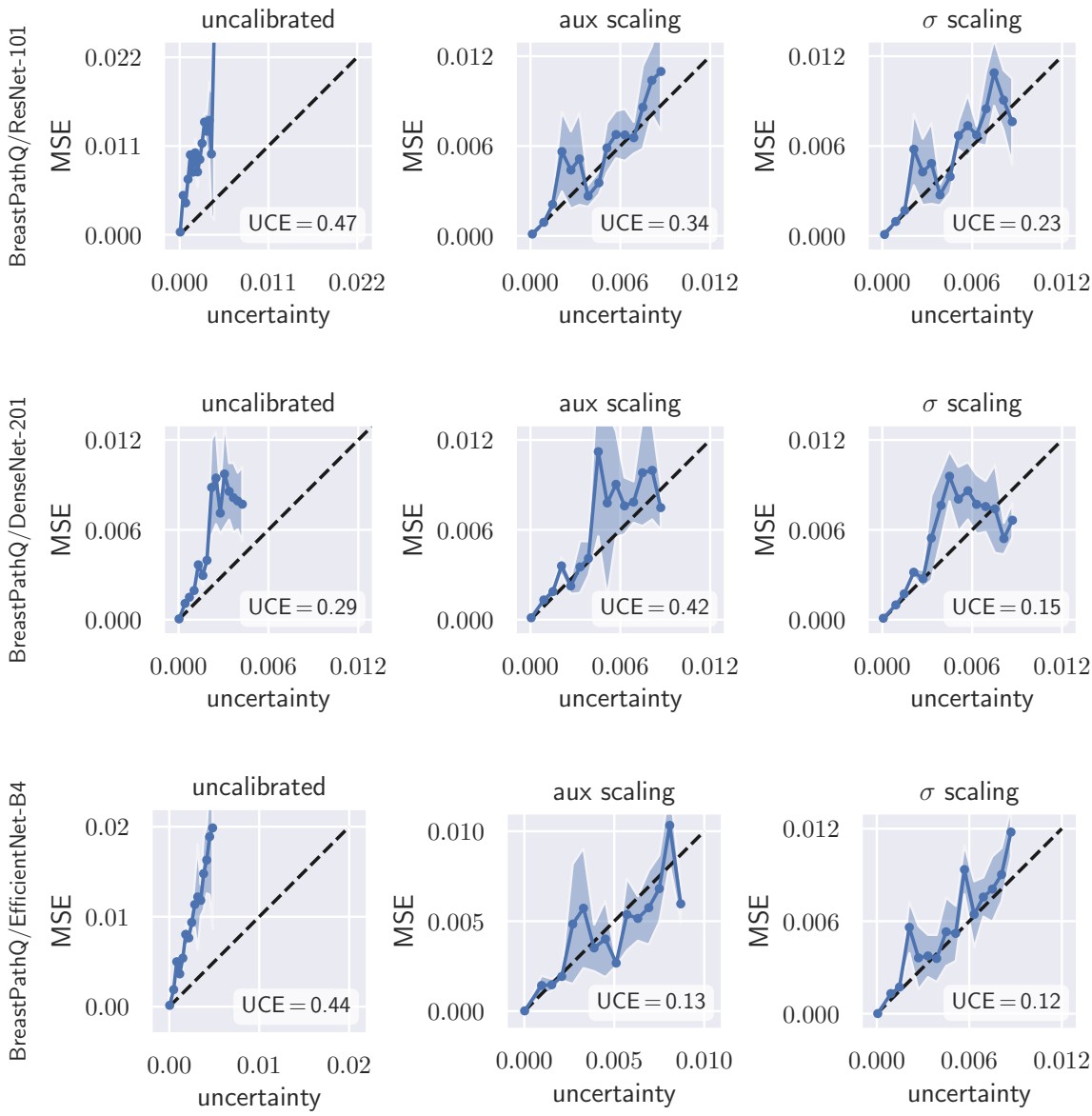

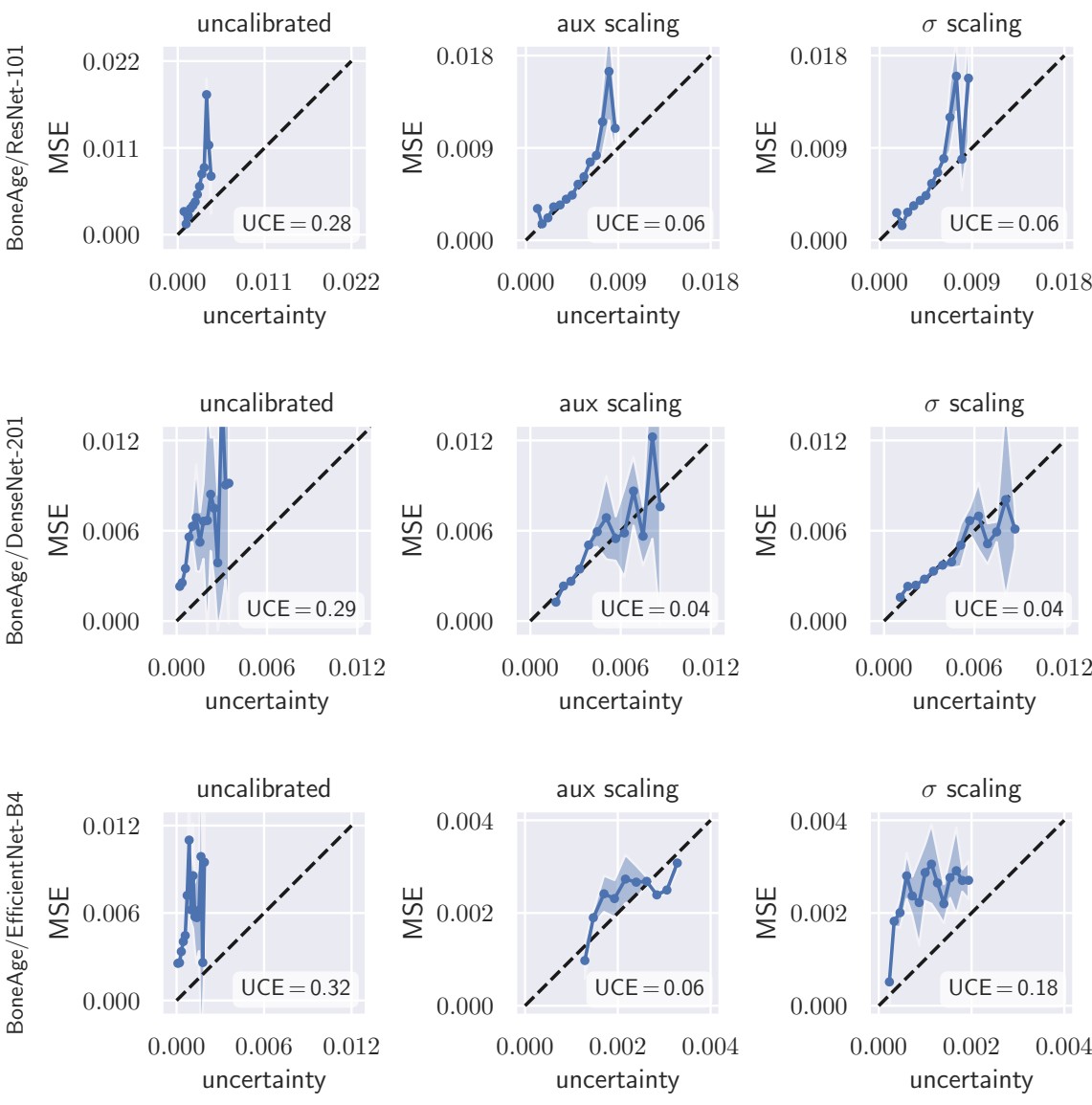

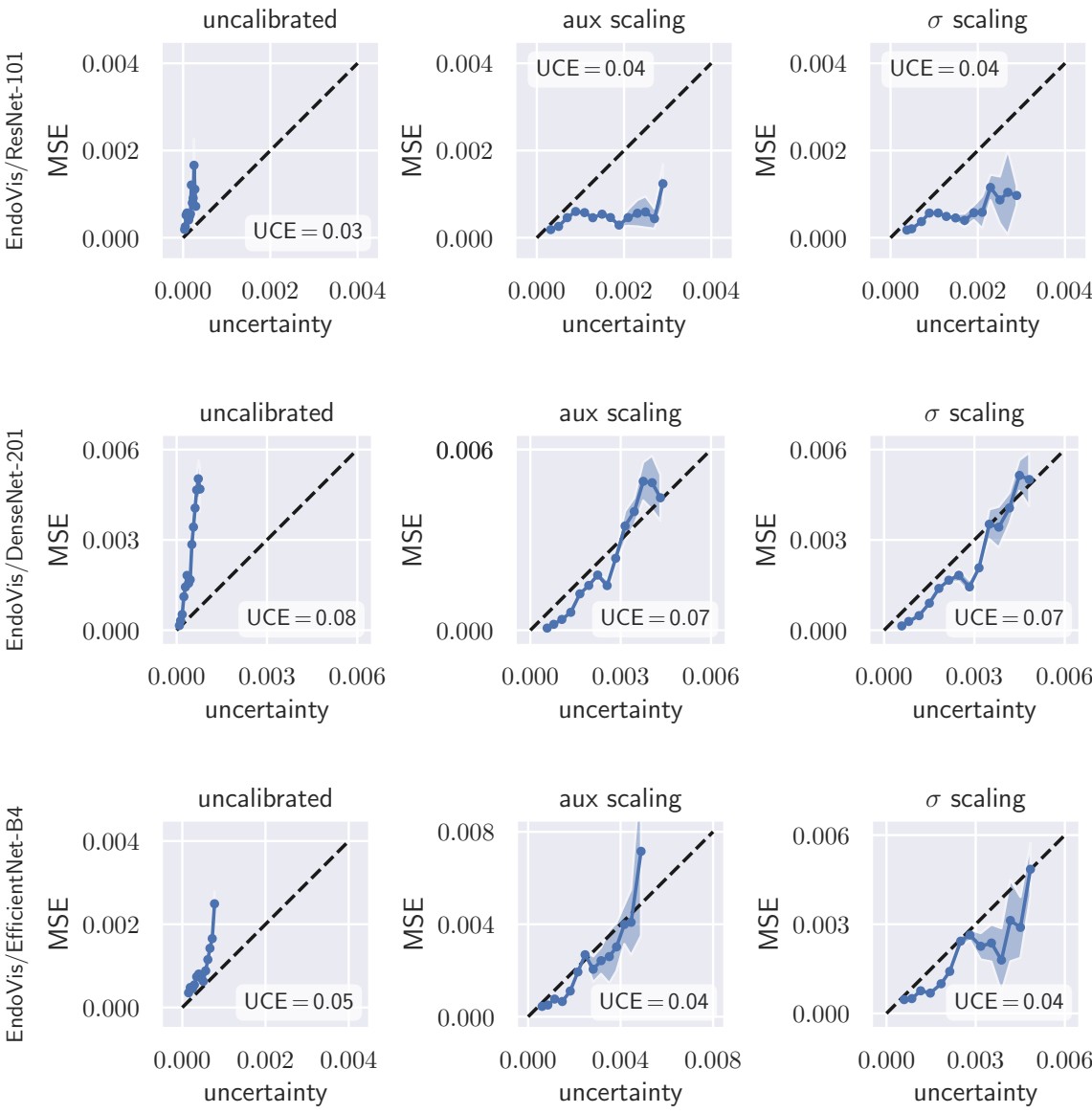

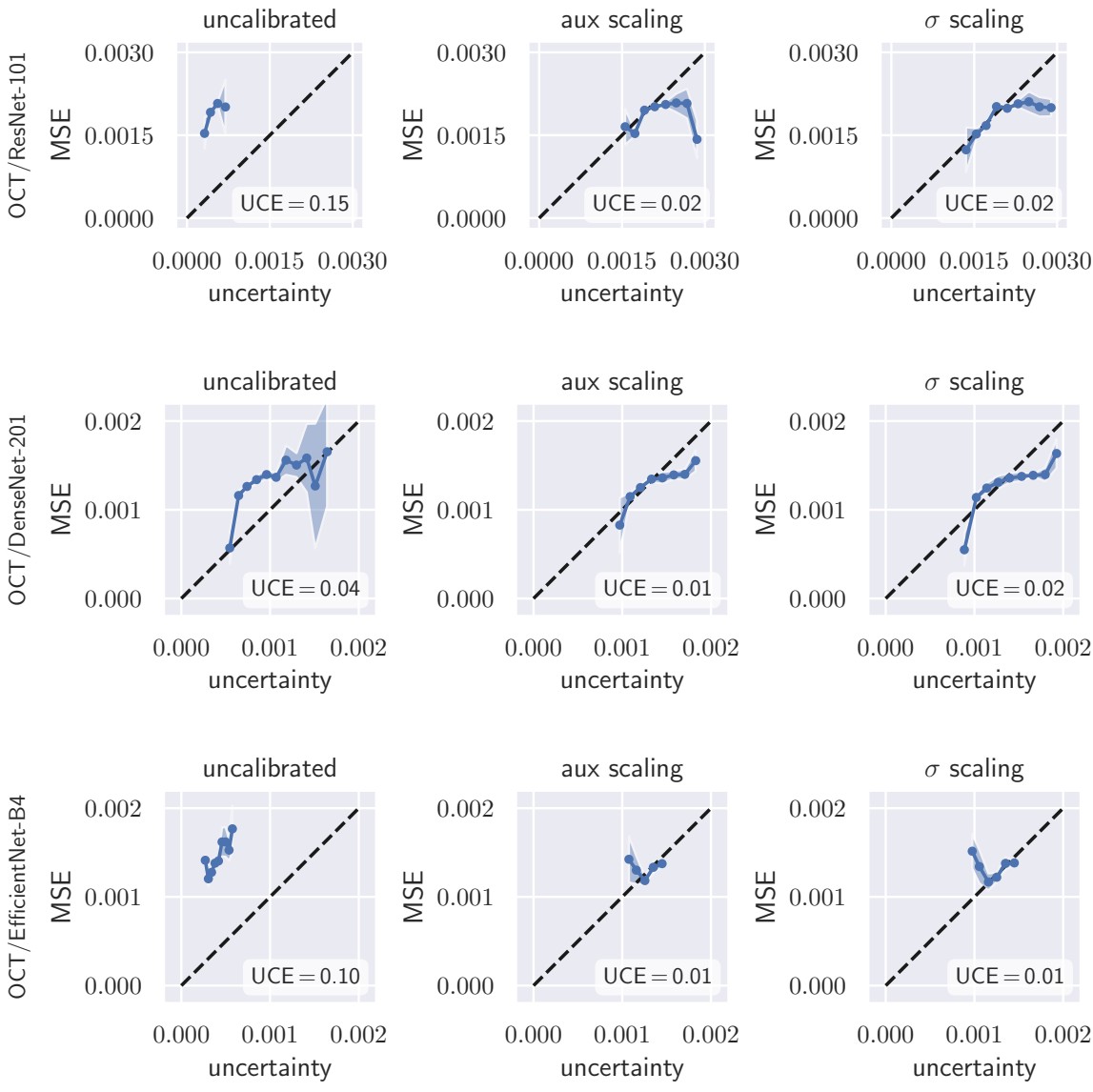

