# OpenReview forum: "Well-Calibrated Regression Uncertainty in Medical Imaging with Deep Learning"
_MIDL.io/2020/Conference — MIDL 2020_

### Official Review · AnonReviewer4 · 2020-03-06
**It is a good contribution to MIDL**

**Rating:** 4
**Confidence:** 5
**Recommendation:** Oral

**Summary:**

This manuscript proposes an optimization framework on top of negative log-likelihood optimization in order to rescale the biased estimation of aleatoric uncertainty. This is an important problem in uncertainty estimation. The presented method is well-evaluated on four different datasets and using several deep architectures and the results show significant improvement over common auxiliary re-scaling. Even though the authors opted to pitch their method around medical applications, the methodological developments are quite general for wild applications.

**Strengths:**

- The paper addresses a relevant and important problem in the application of deep learning on medical imaging.
- The text is clear, the diagrams are descriptive, and the paper is well-organized.
- The authors support the experimental observations with enough analytical expansions.
- The experiments are comprehensive.

**Weaknesses:**

I see only few minor weaknesses in this manuscript:
- The calibration plots are selectively presented. I suggest including the rest of the calibration plots (different datasets and architectures), to the supplementary.
-  Considering using deep learning and the random process involved in the initialization and optimization, all experiments must be repeated at least 10 times and the standard deviations (or confidence intervals) must be reported in Table 1.


**Detailed Comments:**

Some extra minor comments:
- Kuleshov et al. => Kuleshov et al. (2018) use \citet
- Do not use references as nouns. Use \citet instead of \cite.For example: in (Levi et al, 2019) => in Levi et al. (2019)
- in Eq. 5, I guess '+' in the last term should change to '-'. Similarly in Eq. 20 in the supplementary.
- in Eq. 19, in the exponential part a '-' is missing.
- in Section 2.3, the authors explain the procedure for rescaling only the aleatoric uncertainty, while at the end of section 2.4, we suddenly learn that the rescaling is performed on the prediction uncertainty. To keep the text in the right order, it might be more appropriate to swap the texts in sections 2.3 and 2.4 (with some modification in the text of course).


**Justification Of Rating:**

The paper addresses an important problem in the community. The text is clear and well-fitted for didactic purposes. The experimental setups are appropriate and the results are significant. In short, I think this adds a lot to the MIDL.

**Paper Type:**

methodological development

**Questions To Address In The Rebuttal:**

Few questions for more clarification:
- The authors opt to use linear neurons in the last layer for the prediction variance. Considering the fact that variance and standard deviation are always positive, using a positive transfer function (such as softplus) makes more sense here. Please clarify.
- While early-stopping provides a similar strategy to avoid biased estimation, it is not clear why the authors do not compare their method with it. Please clarify.
- The authors used 0.5 for dropout probability in estimating epistemic uncertainty. 0.5 seems a little bit high value for drop out that makes the results unstable. Have you tried smaller values?



**Special Issue:**

yes

---

> ### Author Response · Authors · 2020-04-01
> **Repeated experiments**
>
>
> Dear AnonReviewer4,
> thank you for your positive review. Nevertheless, we try to address your comments as they helped us to improve our manuscript.
>
> Q: "The calibration plots are selectively presented. I suggest including the rest of the calibration plots (different datasets and architectures), to the supplementary."
> A: We will add all calibration plots to the supplementary.
>
> Q: "Considering using deep learning and the random process involved in the initialization and optimization, all experiments must be repeated at least 10 times and the standard deviations (or confidence intervals) must be reported in Table 1."
> A: Thank you for this suggestion. Especially when using a stochastic process such as dropout to obtain uncertainties, repeating the experiments is important. According to your comment, we repeated all test set runs 10 times and report mean and standard deviation of all metrics. We also visualize the standard deviation of each error per bin in the calibration plots as shaded areas.
>
> Q: "The authors opt to use linear neurons in the last layer for the prediction variance. Considering the fact that variance and standard deviation are always positive, using a positive transfer function (such as softplus) makes more sense here. Please clarify."
> A: We implemented our network to directly predict \log(\sigma^2) to avoid numerical instabilities (see answer to AnonReviewer3). For this reason, we do not restrict the output of the network with positive transfer functions. We added a sentence for clarification.
>
> Q: "While early-stopping provides a similar strategy to avoid biased estimation, it is not clear why the authors do not compare their method with it. Please clarify."
> A: Thank you for this suggestion. In hindsight this would be an interesting additional comparison to underline the strength of our approach. We did not compare to early stopping as this would result in considerably higher test set MSE (see Fig. 2).
>
> Q: " The authors used 0.5 for dropout probability in estimating epistemic uncertainty. 0.5 seems a little bit high value for drop out that makes the results unstable. Have you tried smaller values?"
> A: Dropout with p=0.5 is only used for ResNet (as described in Gal, Ghahramani, (2016)). We use the standard dropout p of the architecture (p=0.2 for DenseNet and p=0.4 EfficientNetB4). We changed the misleading statement in Appendix F. We did not observe unstable uncertainty estimations as shown by the standard deviations of the repeated experiments (see Q2 of this rebuttal).
>
> We thank you again for your valuable review. Your remarks helped us a lot.

---

> > ### Comment · AnonReviewer4 · 2020-04-02
> > **Thanks for the clarifications**
> >
> > I am totally satisfied with the authors' response. Given the fact that a portion of the community is unaware of the addressed problem in this paper (miscalibration of estimated uncertainty in deep models), I wish to see this work as a part of MIDL 2020.

---

### Official Review · AnonReviewer1 · 2020-03-12
**Are uncertainty estimates just miscalibrated or flawed?**

**Rating:** 1
**Confidence:** 5

**Summary:**

The paper proposes to correct miscalibrated uncertainty estimates in Deep Learning models, by adjusting the variance in the likelihood model by a scalar factor. The factor is tuned on a hold-out set after a standard training phase.

Preliminary experiments are reported on 4 medical imaging datasets..

**Strengths:**

The paper addresses an important issue in medical imaging.

The paper would be more convincing if it focused on showing that the proposed approach is useful in practice, for instance if estimated confidence values correlate better empirically with predictive error, for the medical application of interest. Maybe the experimental results in the paper could be explained in more detail.

**Weaknesses:**

The paper emphasizes the methodological contribution and positions itself within the Bayesian framework, but this aspect of the work is the most unconvincing.

- Ultimately miscalibration is a bit of a misnomer. The uncertainty estimates correctly reflect the predictive error or they don't. It is unclear why changing flawed estimates by a scalar factor would make them reliable. The proposed method will "hide" the most obvious flaw: the order of magnitude. I would argue that the incorrect order of magnitude is a useful symptom of the flaws in the Bayesian model. Addressing the symptom is unlikely to fix the root cause.

- The Bayesian aspect is emphasized throughout the paper. Why do the authors feel that their work is strongly rooted in Bayesian principles? e.g. use of priors, design and analysis of the model, inference method.

- A major flaw in the analysis stems from how aleatoric uncertainty enters the predictive uncertainty on the regressed variable, when it should not. See below. This puts undue emphasis on the aleatoric uncertainty as a source of miscalibration compared to other model misspecifications (or even the inference method). For instance, depending on the task:
1) model bias (even with NNs) e.g., if important predictors are missing in the input x; or due to choosing 1 out of many possible architectures;
2) the i.i.d. assumption in the likelihood model;
3) lack of proper priors and overfitting.
This is especially relevant for mainstream black-box models whose epistemic uncertainty can vanish in the large data regime due to a string of poor practices (hence the convenience of adding the aleatoric uncertainty).

**Detailed Comments:**

- Eq. 3 is unclear because the dependence of mu and sigma on x is dropped. As a result in the next equations the authors reason as if the values $\mu_\theta^i(x)$ (resp. $\sigma_\theta^i(x)$) were not coupled. [$\mu$'s and $\sigma$'s are coupled as well.]

- In the same vein Eq. 7 and 8. treat each $\mu^i$ and $\sigma^i$ as 1 degree of freedom, which is not the case as the theta's are the real degrees of freedom. This limits the relevance of the analysis.

- "Eq. (6) gives an unbiased estimation of $y^{(i)}$": Unclear statement since $y^{(i)}$ is an observation. Judging from Eq. 6, do you mean an unbiased estimate $\mu_\theta$ of the regressed variable? My impression is that "biased" and "unbiased" refer to many different unrelated things throughout the paper, not always relevant to the main message.

- "$\sigma^2$ is estimated relative to the estimated mean $\mu$ and therefore biased". Acknowledging limitations of MLE/MAP estimation, how does this fit into the overall argument? Isn't the bias introduced by ML estimation rather than a proper Bayesian treatment secondary? The paper does not pursue a Bayesian treatment of sigma anyway.

Assume the true value $\sigma_*$ of the noise level is known. As $m$ grows large and because of the i.i.d. assumption, the predictive variance on $\mu$ will go to $0$; so any bias in the model will cause the uncertainty to be "miscalibrated" anyway.

- Eq. 9, 10, 11 are a direct reparametrization of Eq. 3-6. The scaling $s$ is redundant and, in absence of priors on $\sigma$'s or $s$, does not change the global optimum for $\sigma$ <-> $s \cdot \sigma$. Then, qualitative improvements are due to the two stage training procedure, using a hold-out set to tune $\sigma$. I would argue the procedure is a 'practical' post-processing step that can be explained clearly without the use of Bayesian concepts.

- In Eq. 12, the aleatoric noise should not appear in the expression for the predictive variance. Here, aleatoric noise refers to the process by which observations are corrupted. The predictive variance reflects the uncertainty in the value of the regressed variable (before corruption by measurement).
The confusion is only possible because the regressed variable and the observations are of the same type. What if the regressed variable was the weight (in $\text{kg}$) of a projectile whose position (in meters) is observed, up to sensor noise, at regular time intervals? Clearly the observation noise (variance in $m^2$) would not enter additively into the predictive variance for the regressed weight (in $\text{kg}^2$).

The aleatoric noise only enters the predictive variance indirectly via the epistemic term (higher aleatoric noise generally leads to higher uncertainty on $\theta$). A correct way to retrieve a similar expression to Eq. 12 would be to model the regressed variable $y(x) = NN(x) + b(x)$ as the sum of a model prediction and a model bias $b(x)$, and further assuming that the bias is spatially i.i.d Gaussian. (however that makes the bias unidentifiable in presence of observation noise.)

**Justification Of Rating:**

The main idea in the paper comes down to an ad-hoc correction, but the authors position their work w.r.t. Bayesian uncertainty quantification. From this standpoint, the paper is severely flawed. The work would be more convincing if it focused on strong empirical results, showing that the proposed approach is useful in the medical application of interest, for instance if estimated confidence values correlate better empirically with the predictive error.

**Paper Type:**

methodological development

**Special Issue:**

no

---

> ### Author Response · Authors · 2020-04-02
> **Part 1**
>
>
> Dear AnonReviewer1,
> thank you for your detailed and elaborate review. It has led us to critically question our basic assumptions. We are certain that our assumptions are correct and hope that our response will convince you of this. We tried to address all of your concerns and explain why we think that our method follows the principles of Bayesian deep learning.
>
> Q: "Ultimately miscalibration is a bit of a misnomer. The uncertainty estimates correctly reflect the predictive error or they don't."
> A: We regret that we have not sufficiently described our concept of (mis-)calibration. Our view of "miscalibration" corresponds to the idea of miscalibration as reported by many former publications. A famous definition of calibration can be found in DeGroot et al. (1983): "The concept of calibration pertains to the agreement between a forecaster's prediction and the actual observed relative frequency". Guo et al. addressed miscalibration of confidence (softmax likelihood) for classification (Guo et al., 2017). Our concept of (mis-)calibration is closely related to theirs. An even more consistent definition of miscalibration can be found in § 4.2 of Kendall and Gal, (2017).
> On the basis of the references mentioned above, we are convinced that uncertainty estimates can reflect the predictive error better or worse. In § 2.2. of our paper, we explain why this happens and subsequently show that miscalibration can be corrected by our presented approach. We don't think that our findings are just empirically interesting with respect to the presented application of medical imaging. We hope that you can agree with our concept of (mis-)calibration, that is adapted from the cited literature, as it is a fundamental aspect of our work.
>
> Q: "The Bayesian aspect is emphasized throughout the paper. Why do the authors feel that their work is strongly rooted in Bayesian principles? e.g. use of priors, design and analysis of the model, inference method."
> A: Our understanding of a Bayesian approach to deep learning is very well described in Wilson (2019) and Kendall and Gal, (2017). We place a prior distribution over the neural net's weights (for example a Gaussian N(0, I)) and aim at computing the posterior predictive distribution by marginalization over the parameters. This posterior distribution captures all possible settings of the model's parameters and allows us to reason about the uncertainty of a prediction. Evaluating the posterior predictive distribution is intractable and different approximations to Bayesian inference exist. We have opted for the well-accepted Monte Carlo dropout framework from Gal and Ghahramani (2016).
> According to your question, we added a more detailed explanation to § 2.4 of our manuscript.
>
> Q: "A major flaw in the analysis stems from how aleatoric uncertainty enters the predictive uncertainty on the regressed variable, when it should not. See below. This puts undue emphasis on the aleatoric uncertainty as a source of miscalibration compared to other model misspecifications (or even the inference method)."
> A: We assume that you refer to Eq. (12), which was defined by Kendall and Gal, (2017). In Appendix E we provide mathematical proof that Eq. (12) is an unbiased estimator of the true predictive variance. We regret very much that we do not recognize the flaw of how the aleatoric uncertainty enters the predictive uncertainty. In fact, we expect the epistemic uncertainty to vanish in the theoretical case of an infinitely large training set, and the aleatoric uncertainty to be the main source of uncertainty in this case. In practice, however, data sets are limited and both sources of uncertainty must be taken into account.
>
> Q: "Eq. 3 is unclear because the dependence of mu and sigma on x is dropped."
> A: We have dropped the dependence of mu and sigma on x and added ^{i} as a typographical convenience and regret if this makes our paper more difficult to read. The index \theta denotes that the estimates of \mu and \sigma^2 depend on the neural net's weights. After Eq. (6), we state that \mu and \sigma^2 are estimated jointly. We added the explicit dependence on x for clarification.
>
> Q: "In the same vein Eq. 7 and 8. treat each \mu^{i} and \sigma^{i} as 1 degree of freedom, which is not the case as the theta's are the real degrees of freedom."
> A: Eq. (7) and (8) aims at showing that \hat{\sigma}^{2} is estimated relative to the estimated mean and not relative to the true mean. Its expectation is systematically lower than the true variance and therefore, the estimation of the variance is biased. In our opinion, this still holds even though \mu^{i} and \sigma^{i} both depend on x.

---

> ### Author Response · Authors · 2020-04-02
> **Part 2**
>
>
> Please see the first part of our response below.
>
> Q: "Eq. (6) gives an unbiased estimation of y^{i}": Unclear statement since y^{i} is an observation."
> A: We refer to "unbiased" if the expectation of an estimate equals the true value. In our approach, we assume that, given the input x^{i}, the corresponding value y^{i} has a Gaussian distribution with mean equal to the observation y^{i} (Bishop, 2006). When estimating y^{i} through MLE/MAP, \hat{\mu}^{i} gives an unbiased estimation of y^{i}.
>
> Q: "'\sigma^{2} is estimated relative to the estimated mean \mu and therefore biased'. Acknowledging limitations of MLE/MAP estimation, how does this fit into the overall argument? Isn't the bias introduced by ML estimation rather than a proper Bayesian treatment secondary? The paper does not pursue a Bayesian treatment of sigma anyway."
> A: The bias in the estimation of the aleatoric uncertainty is introduced by ML estimation. However, the Bayesian treatment of the neural net allows us to additionally quantify epistemic uncertainty. The Bayesian treatment alone helps reducing miscalibration. We admit that our paper lacks a proper explanation why the Bayesian treatment sill produces miscalibrated uncertainty and only describe empirical observation. We added a sentence to the outlook to address this shortcoming.
>
> Q: "Assume the true value \sigma_{\ast} of the noise level is known. As m grows large and because of the i.i.d. assumption, the predictive variance on \mu will go to 0; so any bias in the model will cause the uncertainty to be "miscalibrated" anyway. "
> A: If m grows infinitely large, the epistemic part of the predictive variance, which describes the model's ignorance of the data, will go to 0 (Kendall and Gal, 2017; Bishop, 2006). However, the aleatoric uncertainty, i.e., the noise level of the observation process, will not vanish and will be the major source of uncertainty. If m grows, the bias of the estimation of \sigma^2 decreases, which results in a better calibrated model. However, especially in medical imaging, the data sets are limited and incorrect calibration occurs in practice.
>
> Q: "Eq. 9, 10, 11 are a direct reparametrization of Eq. 3-6. The scaling s is redundant and, in absence of priors on \sigma or s, does not change the global optimum for  \sigma<->s \cdot \sigma. Then, qualitative improvements are due to the two stage training procedure, using a hold-out set to tune \sigma. I would argue the procedure is a 'practical' post-processing step that can be explained clearly without the use of Bayesian concepts."
> A: As described above, our Bayesian treatment places a prior onto the model's weights. The rescaling of \sigma with s tackles the under-estimation of \sigma on unseen data. Simply re-training the model in a two stage procedure via Eq. (6) on a hold-out set would result in worse test set MSE and, given enough re-training iterations, would overfit the hold-out set and thus still be miscalibrated. On the other hand, re-scaling only \sigma on a hold-out calibration set helps to fix unter-estimation of \sigma without negatively affecting test set MSE.
>
> Q: "In Eq. 12, the aleatoric noise should not appear in the expression for the predictive variance."
> A: We think that this question was already answered in the response to Q3 of your review.
>
> Thank you again for critically reviewing our submission. We have thoroughly checked our statements and are convinced that they are in agreement with the current and relevant literature in the field of uncertainty quantification in deep learning. We still think that our work constitutes an interesting contribution to the community and hope to be able to further clarify any remaining questions personally at MIDL 2020.
>
> References:
>
> DeGroot, Morris H and Fienberg, Stephen E. The comparison and evaluation of forecasters. The Statistician, pp. 12–22, 1983.
>
> Guo, C., Pleiss, G., Sun, Y., and Weinberger, K. Q. On calibration of modern neural networks. In ICML, pp. 1321–1330, 2017.
>
> Kendall, Alex and Gal, Yarin. What uncertainties do we need in bayesian deep learning for computer vision? In NeurIPS, pages 5574–5584, 2017.
>
> Wilson, Andrew G. The case for Bayesian deep learning. arXiv preprint arXiv:2001.10995., 2019.
>
> Gal, Yarin and Ghahramani, Zoubin. Dropout as a bayesian approximation: Representing model uncertainty in deep learning. In ICML, pages 1050–1059, 2016.
>
> Bishop, Christopher M. Pattern Recognition and Machine Learning. Springer, 2006. ISBN 978-0-387-31073-2.

---

> > ### Comment · AnonReviewer1 · 2020-04-02
> > **Thanks to the Authors for their response and their time, it is appreciated.**
> >
> > At the same time I stand by the content of the review. Here to simplify I only reiterate 1 core issue I have with the analysis in the paper. It is misled by a confusion between the predictive uncertainty on the (regressed) variable of interest vs. the variability on observations. Aleatoric uncertainty enters the latter additively, but this is not so for the former. You refer to Kendall and Gal (2017) with which I am already familiar so to cut to the point, I clarify the argument below using basic facts that can be easily checked from any ML / Bayesian graphical model book e.g., Bishop (2006), Jordan (2004), Barber (2010), Hastie, Tibshirani, Friedman (2017), Koller and Friedman (2009).
> >
> > I go with the definition of aleatoric noise that is provided in the paper:
> > "Aleatoric uncertainty arises from the data directly;  e.g. sensor noise or motion artifacts."
> > "$\hat{\sigma}_\theta$ captures the uncertainty that is inherent in the data (aleatoric uncertainty)"
> >
> > I stick to the paper's notations, although I suspect they are at the root of the problem. The model is put in graphical form as:
> > $x \rightarrow \mu \rightarrow y$
> > where $y$ are the noisy data, and $\mu$ is the (regressed) variable of interest. $\mu$ is noise free and has some deterministic or probabilistic dependence on some input $x$ via some (neural network) model $NN_\theta$. In your case the dependence is deterministic: a given $x$ yields a given $\mu$ (this is not key to anything).
> > $y$ is a noisy observation of $\mu$. This is where the aleatoric noise comes in. The paper uses Gaussian noise $y = \mu + \epsilon$, $\epsilon \sim N(0,\sigma^2)$. $\sigma^2(x)$ is also estimated by the neural network. My actual point is agnostic w.r.t. this choice vs. any other inference method. In fact any other aleatoric noise model $y\sim p(\cdot|\mu,params)$ beyond the standard regression setting would work.
> >
> > The predictive posterior on the variable of interest ($\mu$, not $y$: $y$ is corrupted by measurement noise) for a new input $x$, given previously observed (input,data) pairs $X,Y$, is:
> > $p(\mu|x,X,Y) = \int_\theta p(\mu|x,\theta) p(\theta|Y,X) d\theta$
> > which (as in your work) can be stochastically approximated using $N$ samples from the model, $\theta_n\sim p(\theta|X,Y)$ as:
> > $p(\mu|x,X,Y) \simeq 1/N \sum_n p(\mu|x,\theta_n)$
> > In your case $p(\mu|x,\theta_n)=\delta(\mu,\hat{\mu}_{\theta_n}(x))$ is a point mass describing a deterministic relationship $\hat{\mu}_{\theta_n}(x)=NN_\theta(x)$. This indeed highlights some of the issues, but is not directly relevant to my argument. My argument is that neither $y$ nor the (aleatoric) observation noise level $\sigma^2$ ever directly enter this equation...
> >
> > If we looked at the variability of new noisy measurements $y$ of $\mu$ for input $x$, given training data $X,Y$, we would recover expressions similar to the one you derived, from $p(y|x,X,Y) = \int_\theta p(y|x,\theta) p(\theta|X,Y)d\theta$. However the goal is not to estimate a noisy measurement, it is to estimate the regressed variable.
> >
> > The distinction is crucial because the proposed approach is about re-estimating (a scalar $s$ to recalibrate the noise) $\sigma$ in hopes of improving the uncertainty. The recalibration has a drastic effect for the erroneous expression of predictive uncertainty, but has a very minor effect with the correct expression. Indeed now the measurement noise only enters the predictive uncertainty indirectly via the posterior $p(\theta|X,Y)$ on parameters $\theta$. In the large data regime and due to the i.i.d. assumption on observations, the change is negligible.
> >
> > Beyond dry derivations, I reiterate intuitive remarks as to why aleatoric measurement noise cannot possibly enter additively the predictive variance:
> > 1) Apples and oranges: the unit of the measurement ($y$) vs. that of the variable of interest ($\mu$) can be different in a non-regression setting. Of course the likelihood model $p_\psi(y|\mu)$ would have to be different, but the probabilistic (graphical) structure would be the same, so whatever applies there applies to the present case.
> > 2) A simple 1D regression experiment of a smooth signal corrupted by i.i.d. Gaussian noise is extremely informative as a sanity check. Using suitable priors (e.g. Tikhonov regularisation), and drawing N samples $\mu^1(x_1) \cdots \mu^1(x_k)$, ..., $\mu^N(x_1) \cdots \mu^N(x_k)$, jointly at locations $x_1\cdots x_k$ -- the samples are spatially smooth as expected if not including the aleatoric noise; they are jagged if including the i.i.d. measurement noise, clearly breaking all model assumptions if interpreted as samples from the smooth signal one wishes to estimate.
> >
> > I hope that upon revisiting the review you will find some of it constructive. Best wishes.
> >
> > Kennedy et al. Bayesian calibration of computer models (2001)
> > Arendt et al. Quantification of Model Uncertainty: Calibration, Model Discrepancy, and Identifiability (2012)

---

> > > ### Author Response · Authors · 2020-04-03
> > > **Clarification**
> > >
> > > Dear AnonReviewer1,
> > > we thank you again for your valuable time and the detailed and explanatory response. We have already stretched your patience enough but we try to understand the confusion in the following.
> > >
> > > The goal of our regression model is to predict a target value $y_{\ast}$ given some new input $x_{\ast}$ and a training set of $m$ inputs $ \{ x_1, \ldots, x_m \} $ and their corresponding (observed) target values $ \{ y_1, \ldots , y_m \} $. We assume that $y$ has a Gaussian with mean equal to $ \hat{y}(x) $ and variance $ \hat{\sigma}^{2}(x) $. A Bayesian neural net, approximated with MC dropout, outputs these values for a given input $ f_{\tilde{\theta}}(x) = [\hat{y}, \hat{\sigma}^{2}] $, with model weights drawn from the approximate posterior with $ \tilde{\theta} \sim q(\theta) $.
> > > Here we differ from the notation in our paper ($ \hat{y} $ instead of $ \hat{\mu} $) and follow the notation used in Kendall and Gal, (2017) in the hope of reducing confusion. We find $ \theta $ by minimizing Gaussian NLL on the training set. In inferencing, we are interested in the predictive posterior $ p(y_{\ast} \vert x_{\ast}, X, Y) = \int_{\theta} p(y_{\ast} \vert x_{\ast}, \theta) p(\theta \vert X, Y) d\theta $, which is approximated by $N$ samples from the model.
> > > During inference, we perform stochastic forward passes to sample i.i.d. outputs $ \{ \hat{y}_{\ast}^{(1)}(x_{\ast}) , \ldots , \hat{y}_{\ast}^{(N)}(x_{\ast}) \} $. We obtain our predictive mean $ E[y_{\ast}] = \frac{1}{N} \sum_n \hat{y}_{\ast}^{(n)}(x_{\ast}) $ and predictive variance (or uncertainty) as described in Eq. (12).
> > > We understand why "Eq. (6) gives an unbiased estimation of $y^{(i)}$" is an unclear statement and removed it from the manuscript.
> > >
> > > We observed, given our definition of perfect calibration, that the predictive variance is underestimated and scale it with good empirical results.
> > >
> > > If it helps to solve the confusion, we will rewrite our methods section such that the Bayesian treatment is introduced from the beginning. However, we do not think that this affects the main outcome of our paper.

---

### Official Review · AnonReviewer2 · 2020-03-12
**A method for calibrating predictive uncertainties from deep Bayesian regression networks with experiments suggesting it is effective**

**Rating:** 4
**Confidence:** 4
**Recommendation:** Oral

**Summary:**

The problem of calibrating predictive uncertainties obtained using deep neural networks with Monte Carlo dropout is addressed for regression tasks. A rescaling method involving an optimisation step to find a scaling parameter is proposed. It is evaluated using a modified uncertainty calibration error metric on four datasets, and compared to an auxiliary scaling method.

**Strengths:**

- Addresses an important issue for medical image analysis
- Proposes a sigma-scaling method with theoretical foundation that looks fairly straightforward to implement
- The method improved calibration on the four datasets and three networks reported

**Weaknesses:**

I did not find any major weaknesses.  The empirical evaluation could have been more extensively described but it is acceptable for a method-focused conference paper.

I suggest some minor modifications below.

Moving some of the Appendices material into the main text would improve readability.`

**Detailed Comments:**

- I suggest moving Appendix D into the main text as this is not quite standard and is important to understand the results.
- Appendix C can easily be merged around Table 1 which would be easier for the reader.
- In Section 4 "slightly decrease UCE" should be "slightly increase UCE" I believe.
- It would be good to give the value of s for each case in Table 1 as an extra column.
- Rewrite the Table 1 caption so it describes the content of the Table (including acronym expansions) rather than just discussing the result.

**Justification Of Rating:**

The paper presents a novel method based on sigma-scaling for addressing the important problem of calibrating regression uncertainty.  This problem is particularly relevant in many medical image analysis tasks that need uncertainty measures to inform subsequent processing or decision making. The method could find wide application. Experiments suggest it can work effectively.

**Paper Type:**

methodological development

**Special Issue:**

yes

---

> ### Author Response · Authors · 2020-04-01
> **Improved manuscript**
>
>
> Dear AnonReviewer2,
> thank you for your detailed review.
>
> Q: "I suggest moving Appendix D into the main text as this is not quite standard and is important to understand the results."
> A: Thank you for your suggestion. We moved Appendix D to the main text.
>
> Q: "Appendix C can easily be merged around Table 1 which would be easier for the reader."
> A: According to the suggestions of AnonReviewer3, we added an additional section to address the phenomenon of overfitting the calibration set. We think that the table in Appendix C fits best in this section.
>
> Q: "In Section 4 "slightly decrease UCE" should be "slightly increase UCE" I believe."
> A: You are correct. We changed the text.
>
> Q: "It would be good to give the value of s for each case in Table 1 as an extra column."
> A: We added a column to Table 1 to show values of s accoring to your suggestion.
>
> Q: "Rewrite the Table 1 caption so it describes the content of the Table (including acronym expansions) rather than just discussing the result."
> A: We rewrote the table caption and hope that it now meets your expectations.
>
> Thank you again for your feedback. Your comments helped us to improve our manuscript.

---

> > ### Comment · AnonReviewer2 · 2020-04-02
> > **Satisfied with response**
> >
> > I believe based on the response that my comments can be appropriately addressed in the new version. I would like to see this presented at MIDL.

---

### Official Review · AnonReviewer3 · 2020-03-16
**Interesting Paper discussing relevant topic to MIDL**

**Rating:** 2
**Confidence:** 5

**Summary:**

The paper presents a well-calibrated uncertainty method for regression tasks taking into account both aleatoric and epistemic uncertainties. The method is validated on four different datasets and three different architectures. The core of their method is to train a learnable scalar parameter to rescale the aleatoric uncertainty.

**Strengths:**

- well-written paper
- nice theoretical background on mis-calibrated networks.
- extension of Levi et al 2019
- extensive experiment on four different datasets and three different network architectures.

**Weaknesses:**

- strong claims about robust detection of unreliable predictions and OOD samples, but never shown.
- Modest contribution.
- Missing Comparisons, for example, Lakshminarayanan et al. 2017.
- Lack of in-depth evaluation

Lakshminarayanan, B., Pritzel, A. and Blundell, C., 2017. Simple and scalable predictive uncertainty estimation using deep ensembles. In Advances in neural information processing systems (pp. 6402-6413).

**Detailed Comments:**

Modest Contribution: It is a logical continuation of Laves et al., 2019, who introduced temperature scaling to calibrate the epistemic uncertainty, and Levi et al 2019, who proposed scaling the standard deviation of the aleatoric uncertainty.

Missing Comparison: the proposed method is compared with Kuleshov et al. 2018, however, Deep Ensemble (Lakshminarayanan et al. 2017) has also demonstrated that their uncertainty estimate is well-calibrated for regression tasks. A comparison with this method would enrich the experiment section.

Incorrect Proof: If I understood it correctly, there might be something wrong in your proof (Page 13). The moment you introduced \vec{y} to Eq. 30, the expansion doesn’t hold in Eq. 31. In other words, Eq. (30) != Eq. (31). Thus, your predictive sample variance is not unbiased.

Lack of In-depth Evaluation: The method claims that well-calibrated uncertainty allows rejecting unreliable prediction or detection of OOD samples, however, it was not shown in their evaluation, and has never been discussed. Further, some interesting results were never discussed as well. For example, any reason why the aux. scaling gave worse results than the uncalibrated ones, e.g. all network arch. of EndoVis. Also, I was expecting to see Fig.2 after calibration.

**Justification Of Rating:**

I really liked the paper and the topic is definitely relevant, and of high-importance, to Medical Imaging. However, there are some issues have to be fixed before accepting the paper. Most importantly, the comparison with Ensemble Networks, and Lack of in-depth evaluation and discussion.


**Paper Type:**

both

**Questions To Address In The Rebuttal:**

Questions:

1. Can’t we write the solution of Eq. (11) in close form solution? For example, s = +/- \sqrt\big( \sum_{i=1}^m (\sigma_{\theta}^{(i)})^{-2} \| \vec{y}^{(i)} - \hat{\mu}^{(i)}_{\theta}\|^2 \big). If so, then wouldn't the scaling mainly depends on dominator, i.e. the size of your validation set? In other words, if your validation set is too big, then s would be almost zero, which is not desirable then, right?

2. The formula in Eq. (6) might result in numeral instability, do the authors implement in practice similar to Kendall and Gal et al. 2017? If not, how they avoid any numerical stability, in particular, for overfitted models?

3. I was expecting to see more practical, clinically relevant, examples. The one presented in Fig.4 is good, but the model is going good job as well. I would like to see examples showing unreliable predictions, or OOD examples as the authors claim in their abstract and conclusions.


**Special Issue:**

no

---

> ### Author Response · Authors · 2020-04-01
> **Added experiments**
>
>
> Dear AnonReviewer3,
> thank you for your thorough review. It is of high value and allows us to improve our paper substantially. In the following we will address all points raised and hope that we will meet your requirements.
>
> Q: "1. Can’t we write the solution of Eq. (11) in close form solution?..."
> A: Thank you for pointing out this interesting fact. It is indeed possible to write the solution of Eq. (11) in closed form. Based on your suggestion, we added it to the manuscript and changed the surrounding text accordingly. To the best of our knowledge, we need to add an additional factor 1/m as well and came to the conclusion, that s does not vanish with increasing calibration set size. In our experiments, this solution provides the same result as using SGD.
>
> Q: "2. The formula in Eq. (6) might result in numeral instability, do the authors implement in practice similar to Kendall and Gal et al. 2017?"
> A: We did implement Eq. (6) in similar practice to Kendall and Gal 2017 to avoid division by zero:
>
> def nll_criterion_gaussian(mu, logvar, target, reduction='mean'):
>     loss = torch.exp(-logvar) * torch.pow(target-mu, 2).mean(dim=1, keepdim=True) + logvar
>     return loss.mean() if reduction == 'mean' else loss.sum()
>
> The CNNs are trained to directly predict log(\sigma^2). We initially left this fact out of the manuscript, but now we see how important it is to mention it. We added a sentence for clarification.
>
> Q: "3. I was expecting to see more practical, clinically relevant, examples. [...] I would like to see examples showing unreliable predictions, or OOD examples as the authors claim in their abstract and conclusions. "
> A: Three of the four data sets employed were part of public challenges within the medical imaging community. We have assumed that the data sets have sufficient clinical relevance. We regret that the examples we have selected do not meet your expectations.
> We have taken your review as an opportunity to conduct another experiment. We trained a 5-ensemble on the BoneAge data set as described in Lakshminarayanan et al. 2017 and compare it to our approach with regard to quality of uncertainty and OOD detection. We added § 3.1 to describe the experimental setup and results.
>
> Q: "Incorrect Proof: If I understood it correctly, there might be something wrong in your proof (Page 13). The moment you introduced \vec{y} to Eq. 30, the expansion doesn’t hold in Eq. 31. In other words, Eq. (30) != Eq. (31)."
> A: We thank you for also carefully reviewing the appendix of our manuscript. We admit that the expansion from Eq. (30) to (31) is not obvious. We added additional steps to the proof to clarify that Eq. (30) == Eq. (31).
>
> Q: "Further, some interesting results were never discussed as well. For example, any reason why the aux. scaling gave worse results than the uncalibrated ones, e.g. all network arch. of EndoVis"
> A: Thank you for pointing out that we have not discussed some of the results in sufficient detail. We observe that the more powerful aux. scaling sometimes overfits the calibration set. On EndoVis, where the uncalibrated model is not as miscalibrated as on other data sets, this can lead to worse calibration. We moved Appendix C to the main manuscript and added a section to the discussion to address this phenomenon properly.
> The reason why aux. scaling sometimes has worse UCE but better NLL was our definition of perfect calibration. Training with NLL of a Gaussian model favors that the MSE is equal to the variance. In our definition of perfect calibration, the squares were missing, and we have adapted it to this fact, see Eq. (14). We are convinced that this definition better reflects the quality of calibration. This has a slight effect on the UCE values, but the main finding of our work remains unchanged.
>
> Q: "Also, I was expecting to see Fig.2 after calibration."
> A: For the submitted manuscript, we performed re-calibration after the training procedure (after 500 epochs). To comply with your suggestion, we repeated the training for the models shown in Fig. 2 and performed a calibration after each epoch. We have added a corresponding figure to the results section.
>
> We hope that our revision meets your expectations. Your comments have greatly helped us to increase the quality of our work.

---

### Meta-Review · Area_Chair1 · 2020-04-06
**MetaReview of Paper212 by AreaChair1**

**Rating:** 3
**Recommendation For Accepted Papers:** Poster

**Metareview:**

All reviewers highlight the importance of the topic being adressed and 3 out of 4 reviewers agree that the paper introduces an interesting treatment of the problem with convincing evaluation results. The main criticism from the remaining reviewer pertains to the methodological treatment of the graphical model. I feel these comments are fair and already led to interesting discussion with the authors who agreed on going for a more rigorous treatment in a revised version of the manuscript. Overall, I feel the paper will lead to interesting discussions at the conference and includes sufficient material for presentation at the conference.

**Paper Type:**

methodological development

**Special Issue:**

no

---

> ### Author Response · Authors · 2020-04-20
> **Thanks to all reviewers and the area chair**
>
> We welcome the area chair's decision and once again thank all reviewers for their valuable feedback and good discussion. We look forward to stimulating discussions and a great exchange of ideas at this year's MIDL.

---

### Decision · Program_Chairs · 2020-04-11

Accept